# LEARNING INTERPRETABLE CONTROL INPUTS AND DYNAMICS UNDERLYING ANIMAL LOCOMOTION

**Thomas Soares Mullen**[1][*], **Marine Schimel**[2][*], **Guillaume Hennequin**[2], **Christian K. Machens**[1]
**Michael B. Orger**[1][✉]&**Adrien Jouary**[1][✉]

[1] Department of Neuroscience, Champalimaud Foundation
[2] Department of Engineering, University of Cambridge

[✉]{adrien.jouary,michael.orger}@research.fchampalimaud.org

## ABSTRACT

A central objective in neuroscience is to understand how the brain orchestrates movement. Recent advances in automated tracking technologies have made it possible to document behavior with unprecedented temporal resolution and scale, generating rich datasets that can be exploited to gain insights into the neural control of movement. One common approach is to identify stereotypical motor primitives using cluster analysis. However, this categorical description can limit our ability to model the effect of more continuous control schemes. Here, we take a control-theoretic approach to behavioral modeling and argue that movements can be understood as the output of a controlled dynamical system. Previously, models of movement dynamics, trained solely on behavioral data, have been effective in reproducing observed features of neural activity. These models addressed specific scenarios where animals were trained to execute particular movements upon receiving a prompt. In this study, we extend this approach to analyze the full natural locomotor repertoire of an animal: the zebrafish larva. Our findings demonstrate that this repertoire can be effectively generated through a sparse control signal driving a latent Recurrent Neural Network (RNN). Our model's learned latent space preserves key kinematic features and disentangles different categories of movements. To further interpret the latent dynamics, we used balanced model reduction to yield a simplified model. Lastly, we demonstrate the flexibility of our model by successfully applying it to the study of continuous locomotion in another organism, C. elegans. Collectively, our methods serve as a case study for interpretable system identification, and offer a novel framework for understanding neural activity in relation to movement.

## 1 INTRODUCTION

Over the past decade, several techniques have emerged to facilitate the high-throughput, automated tracking of animal behavior (Pereira et al., 2020). This capability is crucial for various applications, from understanding genetics and neural activity (Pereira et al., 2020; Krakauer et al., 2017) to testing the effects of new pharmaceutical compounds (Wiltschko et al., 2020).

A commonly used approach to quantifying animal behavior has been to segment postural time series into a series of discrete actions. This approach has been used effectively to represent the behavioral repertoires of species such as Drosophila (Berman et al., 2014), zebrafish (Marques et al., 2018), and mice (Wiltschko et al., 2015). However, this discretization can obscure continuous variations in motor patterns and does not provide insights directly into the mechanisms that generate these actions. Instead of compartmentalizing behaviors into distinct categories, we suggest a novel approach: treat these observable movements as the output of an underlying latent dynamical system related to motor function. This perspective reframes the study of motor actions as a system identification problem, aiming to uncover the latent control signals and dynamics that govern an animal's full range of movements.

---

[*]These authors contributed to this work.

Prior work (Sussillo et al., 2015; Saxena et al., 2022) has shown that RNNs trained to replicate muscle activation in simple tasks, such as reaching towards a target, closely mimic features of concurrent brain activity. We aim to extend this line of work to natural behaviors, where the animal's actions are not constrained to a specific task, and the cues for movement may be unobserved or internally generated. To achieve this, we propose to fit a latent dynamical model that captures the movement repertoire of an animal, and, at the same time, to infer the unknown control signals that instigate or modify actions.

Here, our goal is thus to identify both the latent control signals and the underlying dynamics that make up the complete locomotor repertoire of an animal. We use iLQR-VAE (Schimel et al., 2022) to learn the latent RNN and infer the control signals from time series of behavioral observations. While we find that more expressive and overparametrized RNNs are easier to learn and provide better fits of the data, even when they are considerably higher-dimensional than the ground truth (Sedler et al., 2022), we additionally propose a scheme to reduce our models to a minimal form, that is easier to dissect and relate to behavior. More specifically, we turn our focus to linear models of the data, and use balanced model reduction (Moore, 1981) to recover the minimal dynamical system that can still capture the system's input-output response.

In the following, we first apply this framework to a system where rich behavioral data sets are available: the zebrafish. This species is well suited to test our approach because its movement repertoire has already been systematically characterized (Marques et al., 2018; Mearns et al., 2020; Johnson et al., 2020), offering a useful benchmark. We train a family of RNNs and show that sparse control signals activated at the onset of actions are sufficient for reproducing the observed postural sequences. These sparse control signals form a compressed representation that encompasses both discrete and continuous descriptions of movement. In the latent space, the trajectories associated with different movements are efficiently "untangled". Notably, this representation also yields spatial navigation information that was not explicitly provided in the training data. Finally, we show that the learned high-dimensional dynamics can be systematically reduced to a low-order, interpretable linear model that reveals the principal modes underlying behavior. Next, we apply the method to modeling continuous locomotion in C. elegans. There, sparse inferred control signals successfully match external stimuli applied to the animal, and model reduction reveals different patterns of crawling and coiling.

We believe our investigation provides a novel framework for studying movements, that will facilitate the understanding of neural activity that generates them, as well as aiding in the investigation of perturbations induced by pharmaceutical or optogenetic treatments. Furthermore, our work serves as a valuable case study for interpretable system identification.

## 2 RELATED WORK

Below, we review existing methods that have been proposed to uncover the dynamics underlying behavior, before detailing how our approach differs from previous work.

**Switching Linear Dynamical Systems** In the context of behavioral data modeling, switching Linear Dynamical Systems (LDS) models have been particularly popular, and are at the core of some of the most widely used algorithms for behavioral segmentation (Datta, 2019). These models assume the existence of a hidden, discrete state that parameterizes the data-generating process, with different dynamics being used depending on the value of the hidden state. MoSeq (Datta, 2019) employs an autoregressive generative process coupled with a sticky Hidden Markov Model. An alternative strategy employs adaptive segmentation algorithms that rely on statistical model testing instead of explicit state transitions (Costa et al., 2019). Such models have been used to successfully segment behavioral time series in various contexts (Johnson et al., 2016; Batty et al., 2019), and relating switches in behavior to concurrent neural recordings has yielded exciting insights into the neural control of behavior (Markowitz et al., 2023). However, a potential limitation of switching LDS models is their reliance on multiple underlying dynamical systems, which complicates the interpretation of the learned dynamics.

**Input-driven latent dynamics** Another way of modeling behavioral dynamics while accounting for potential discontinuities due e.g. to descending commands from the brain, is to use *input-driven* latent dynamical systems models. However, simultaneously learning a system's dynamics and any

unobserved input drive is a very challenging, and often degenerate problem. Two methods, LFADS (Pandarinath et al., 2018) and iLQR-VAE (Schimel et al., 2022), have recently been proposed and shown to give promising results on various synthetic and real datasets. These two methods formulate the learning problem as a variational autoencoder (VAE; Kingma & Welling, 2013) and approximate the posterior distribution over unobserved inputs with a Gaussian distribution. They differ critically in the way the posterior over inputs is computed: in LFADS, the posterior mean is amortized using a bidirectional recurrent neural network (bi-RNN), while iLQR-VAE uses iLQR (Li & Todorov, 2004), a nonlinear control algorithm, to obtain the inputs most likely to have generated the observations given the model parameters. Additionally, LFADS has only been used with a choice of autoregressive prior over the inputs, while iLQR-VAE has been shown to work with a *sparse prior*. While there have been very few examples of work modeling behavioral observations using such input-driven, latent dynamical models, we note that Wimalasena et al. (2022) used AutoLFADS (Keshtkaran et al., 2022) to model muscle activity in mice and monkeys. Importantly however, they made the assumption that the muscles were driven by continuous, autoregressive inputs. The approach we present below differs from this by enforcing a sparse prior over the inputs, which encourages autonomy in the dynamics, and leads to a more interpretable control signal.

## 3 METHODS

### 3.1 SYSTEM IDENTIFICATION USING ILQR-VAE

In this work, we modeled time series of behavioral observations $o$ as the output of a latent dynamical system whose state $z$ is being driven by a set of control inputs $u$, i.e. as :

$$\mathbf{z}_{t+1}^{(k)} = f_\theta(\boldsymbol{z}_t^{(k)}, \boldsymbol{u}_t^{(k)}) \qquad (1) \qquad \mathbf{o}_t^{(k)} \sim \mathcal{N}(\boldsymbol{C}\mathbf{z}_t^{(k)}, \boldsymbol{\Sigma}). \qquad (2)$$

In Equation (1), $\mathbf{o}_t^{(k)} \in \mathbb{R}^{n_0}$ is the temporal evolution of the behavior in the $k$-th set of observations, $\mathbf{z}_t^{(k)} \in \mathbb{R}^n$ is a latent representation of that behavior, and $\mathbf{u}_t^{(k)} \in \mathbb{R}^m$ are the control inputs driving the latent dynamics. Note that we used inputs to set the initial state of the dynamics in the latent space, that is, $\mathbf{z}_1 = f_\theta(\mathbf{0}, \mathbf{u}_0, 0)$. Therefore, given a set of parameters $\theta$, the input sequence fully determines the latent trajectory $\mathbf{z}(\mathbf{u}) = \{\mathbf{z}_1, \mathbf{z}_2, \ldots, \mathbf{z}_T\}$, and thus the corresponding output. In this model, inputs therefore represent the true latent variables of interest. As behavioral datasets often consist of real-valued observations, we assumed Gaussian observation noise throughout our work, although other noise models could be used in other settings.

Learning the dynamics of a system driven by unobserved inputs is a challenging system identification problem. We tackled it using the recently proposed iLQR-VAE method (Schimel et al., 2022, see Appendix A) which optimizes a lower bound on the log marginal likelihood w.r.t. the model parameters $\theta$ (which include any parameters in $f_\theta(\cdot)$, $\mathbf{C}$, $\boldsymbol{\Sigma}$, and the prior parameters; see below). To facilitate identifiability, here we assumed that the dynamics were driven by a *sparse* set of inputs. Specifically, we followed Schimel et al. (2022) and used a Student-t prior distribution (see details in Appendix B.1). Because of its heavy-tailedness, this choice of prior is consistent with the model using small inputs most of the time, whilst still accommodating the occasional large input. It is thus well suited to inferring large sporadic inputs that may trigger e.g. switches in behavior, while encouraging the model to capture most behavioral segments via strong, near-automonous dynamics.

In iLQR-VAE, the posteriors over inputs are obtained using iLQR for each set of contiguous observations; inference can be performed on any new time series not seen during training, even if its length is different from the training data. After training, the model described above thus provides us with a) a set of parameters $\theta$ that can be dissected further, and b) a posterior distribution over the inputs, for every set of observations. Assuming the behavior can be generated with sparse, low-dimensional inputs, the posterior over inputs $p_\theta(\mathbf{u}|\mathbf{o})$ yields a compressed representation of the high-dimensional time series, which is typically more interpretable and easier to visualize than the original data.

### 3.2 MINIMAL INTERPRETABLE DYNAMICS USING BALANCED MODEL REDUCTION

Alongside nonlinear RNNs, we also fitted the data using linear dynamics, i.e modeled $f_\theta$ in Equation (1) as

$$f_\theta(\boldsymbol{z}_t^{(k)}, \boldsymbol{u}_t^{(k)}) = \boldsymbol{A}\boldsymbol{z}_t^{(k)} + \boldsymbol{B}\boldsymbol{u}_t^{(k)}. \qquad (3)$$

The linear model is characterized by its parameters ($\boldsymbol{A} \in \mathbb{R}^{n \times n}$, $\boldsymbol{B} \in \mathbb{R}^{n \times m}$, $\boldsymbol{C} \in \mathbb{R}^{n_o \times n}$). However, the smallest dimension $n$ required to accurately model the dynamics is unknown. One way of determining the appropriate size of the dynamical system involves fitting models of various sizes and comparing their generalization performance on test data. Here, we instead propose to fit a high-dimensional, potentially overparametrized linear model, and use *balanced truncation* (Moore, 1981; Antoulas, 2005) post-hoc to obtain a reduced-order model that still accurately captures the transfer function of the original model.

Balanced truncation is performed in two stages; in a first step, the original dynamics ($\boldsymbol{A}$, $\boldsymbol{B}$, $\boldsymbol{C}$) are transformed into an equivalent balanced system ($\tilde{\boldsymbol{A}}$, $\tilde{\boldsymbol{B}}$, $\tilde{\boldsymbol{C}}$), i.e. a dynamical system with the same transfer function as the original model, but in which each mode is as controllable as it is observable. In a second stage, this balanced realization of the original system is truncated to a desired order $r < n$. Balancing the system in step 1 relies on computing its observability and controllability Gramians. Those are positive-definite matrices defined respectively as

$$\boldsymbol{W}_o = \int_0^\infty e^{\boldsymbol{A}^T t} \boldsymbol{C}^T \boldsymbol{C} e^{\boldsymbol{A} t} dt, \quad (4) \qquad \boldsymbol{W}_c = \int_0^\infty e^{\boldsymbol{A} t} \boldsymbol{B} \boldsymbol{B}^T e^{\boldsymbol{A}^T t} dt. \quad (5)$$

They capture the information about the state space directions in which the system is most sensitive to inputs, and those in which it is most likely to elicit outputs. More precisely, the top eigenmodes of $\boldsymbol{W}_o$ correspond to the most observable directions of the system, i.e. directions which will lead to most variance in the output when the network is initialized along them. The top eigenmodes of $\boldsymbol{W}_c$ are the most controllable directions of the linear system, i.e correspond to directions along which the dynamics can be steered at a minimal input energy cost (Kao & Hennequin, 2019). Importantly, given ($\boldsymbol{A}$, $\boldsymbol{B}$, $\boldsymbol{C}$), $\boldsymbol{W}_o$ and $\boldsymbol{W}_c$ can be computed in closed form as the solutions to two dual Lyapunov equations. Finding a balanced realization corresponds to finding a similarity transformation $\boldsymbol{T}$ such that, in the transformed system characterized by parameters $\{\tilde{\boldsymbol{A}} = \boldsymbol{T}^{-1} \boldsymbol{A} \boldsymbol{T}, \tilde{\boldsymbol{B}} = \boldsymbol{T}^{-1} \boldsymbol{B}, \tilde{\boldsymbol{C}} = \boldsymbol{C} \boldsymbol{T}\}$, the resulting Gramians $\tilde{\boldsymbol{W}}_o = \boldsymbol{T}^T \boldsymbol{W}_o \boldsymbol{T}$ and $\tilde{\boldsymbol{W}}_c = \boldsymbol{T}^{-1} \boldsymbol{W}_c (\boldsymbol{T}^{-1})^T$ become equal to each other (balanced) and diagonal. In other words,

$$\tilde{\boldsymbol{W}}_o = \tilde{\boldsymbol{W}}_c = \boldsymbol{\Sigma}^2 = \text{diag}(\sigma_1^2, \sigma_2^2, \ldots, \sigma_n^2). \quad (6)$$

In Equation (6), $\sigma_i, i = 1, \ldots, n$ are the ordered, positive Hankel singular values (HSVs) of the system. These play the same role for the dynamical system as singular values do for constant, finite-size matrices. Thus, in the same way as the decay of the singular values can be used to choose the dimensionality cutoff in principal component analysis (PCA), evaluating the HSV spectrum provides a principled way of deciding the truncation dimension $r$. Once $r$ has been set, truncation of the system is achieved by selecting the first $r \times r$ block of $\tilde{\boldsymbol{A}}$, $\tilde{\boldsymbol{A}}^{(r)}$, the first $r$ rows of $\tilde{\boldsymbol{B}}$, $\tilde{\boldsymbol{B}}^{(r)}$, and the first $r$ columns of $\tilde{\boldsymbol{C}}$, $\tilde{\boldsymbol{C}}^{(r)}$. The dynamics of the reduced dynamical system can then be summarized as follows:

$$\tilde{\boldsymbol{z}}_{t+1}^{(k)} = f_\theta(\tilde{\boldsymbol{z}}_t^{(k)}, \boldsymbol{u}_t^{(k)}) = \tilde{\boldsymbol{A}}^{(r)} \tilde{\boldsymbol{z}}_t^{(k)} + \tilde{\boldsymbol{B}}^{(r)} \boldsymbol{u}_t^{(k)} \quad (7)$$

The removal of dimensions that are uncontrollable and unobservable provides a simple, yet effective, way to reduce the latent dynamics of the system to a minimal form that can be interpreted and analyzed with more ease.

As a proof of concept, we validated the balanced model truncation's ability to recover the true dynamics of a system using a synthetic example. In Appendix C, we generated data from a set of lightly damped pendulums coupled by a spring and driven by sparse forcing inputs. We modeled the pendulum dynamics as linear, and fitted a 20-dimensional latent linear dynamical system to the data. The spectrum of the Hankel singular values exhibited a drop after 4 modes corresponding to the true dimensionality of the data. Finally, truncating the dynamical system to 4 dimensions successfully revealed the modes corresponding to the underlying synthetic dataset (see Figure S1). This validation emphasizes how combining the iLQR-VAE with balanced model reduction can yield a minimal latent dynamical system that accurately describes the data.

## 4 RESULTS

### 4.1 MODELING ZEBRAFISH LOCOMOTION

Next, we turned to modeling a dataset of behavioral observations in larval zebrafish. This case study highlights how our modeling approach enables us to (i) gain insights into the latent space underlying

this complex behavior, and (ii) extract a minimal model of the underlying dynamics, which is more amenable to further analyses.

### 4.1.1 ZEBRAFISH BEHAVIORAL DATASET

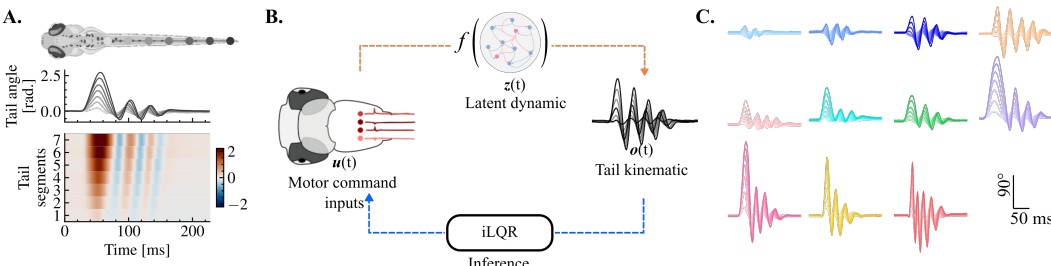

Figure 1: **Overview of the model setup for zebrafish Locomotion** (A) Tail angle tracked along 7 tail segments (top). Example swim bout captured by displacement of tail angles along tail segments through time displays a wave propagating along the tail. (B) Illustration of the model setup. We learn a generative model from inputs to observations, and use iLQR-VAE to perform inference in this model. (C) Illustration of the behavioral repertoire, one movement is shown for each category.

The larval zebrafish is a model organism studied in thousands of biological research labs, owing to its optical transparency, rapid developmental cycle, and the extensive genetic library available (Mrinalini et al., 2022). Zebrafish larvae swim using discrete episodes of propulsion, known as swim bouts, which typically last around 200ms. The body shape of the zebrafish can be characterized by measuring the bending angle as a function of distance along the length of its body (Figure 1A). We compiled a dataset of 30800 tail angle times series classified according to the known repertoire of zebrafish larva (Marques et al. (2018), details in Appendix G.

### 4.1.2 LOW-DIMENSIONAL, SPARSE CONTROL OF LARVAL ZEBRAFISH LOCOMOTION

To characterize zebrafish locomotion, we sought to learn dynamical systems that were able to generate its locomotor repertoire (see Figure 1B).

**Model selection** To find the models that best described the data, we varied the dimensions of the control signal and of the latent size, as well as the RNN architecture. We considered both MGU RNNs (Zhou et al., 2016; details in Appendix B.2), and Linear Dynamical Systems (LDS). We evaluated models based on both (i) their ability to reconstruct the data (fraction $R^2$ of variance explained by the mean of the posterior predictive distribution over observations), and (ii) the sparsity of the inferred control signals. We quantified temporal sparsity by the ratio of the $L^1$-norm to the $L^\infty$-norm. Therefore, a sparsity score of 1 corresponds to an input signal that is non-zero at a single time point, resembling an impulse function. We found that all trained models could generate accurate reconstructions of our dataset ($R^2 > 0.94$; Table S1). The output signal predicted by the dynamical system's flow was smoother than the behavioral observation, i.e. our method acted as a denoising autoencoder. The high reconstruction fidelity can be attributed to the inherently low-dimensional nature of behavioral recordings. Yet, a critical concern was ensuring the control signal captured the genuine dynamics rather than simply mirroring zebrafish posture. Thus, we also evaluated models on the sparsity of control. We found that matching the observations *using a very sparse control* was challenging, and possible only for the largest and most expressive models. Both an LDS and an MGU network of latent size $n = 120$ driven by $m = 10$ inputs could accurately reconstruct the zebrafish behavior while using sparse inputs (Figure 2A). As we detail below, however, the MGU network was able to compress the behavioral information more effectively. For our dissection of the latent controls driving the system, we thus turned our focus to this network.

**Sparse control signal and latent state representation** Our assumption regarding the sparsity of the control signal was grounded in biological observations. Indeed, sparse electrical activation of brainstem neurons projecting to the spinal cord has been shown to be sufficient to elicit forward locomotion or escape (Severi et al., 2014; Xu et al., 2021). In addition, Markov et al. (2021) established that zebrafish bouts are modified by unexpected visual feedback only after 220ms, with the intial portion of the bout proceeding in a ballistic fashion after the onset.

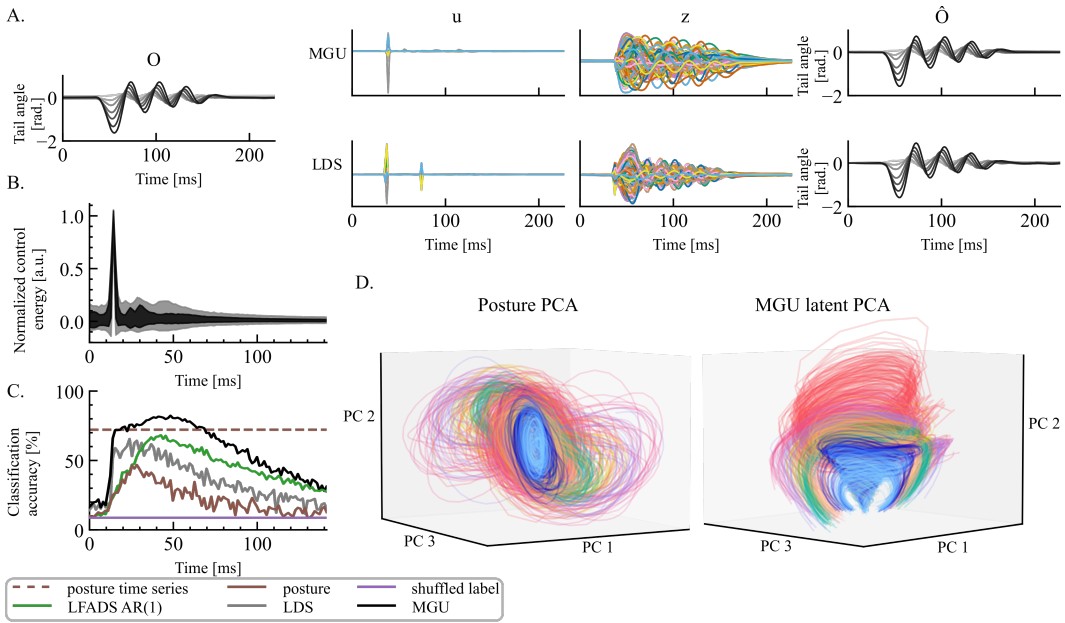

Figure 2: **Sparse inferred inputs generate a range of swim bouts**. (A) Example bout reconstruction ($R^2 = 0.97$ between the observation $O$ and the reconstruction $\hat{O}$), with a sparse input $u$ driving the latent trajectories $z$ for MGU and LDS (B) Mean $\pm$ standard deviation across dataset of the normalized $L^2$-norm of $u$ aligned on the first peak. MGU RNN (black) learns a more transient input to generate movements across the dataset compared to LDS (gray). The alignment to the first peak is the same is B., C. and D. (C) Performance of linear classifier to predict the movement category using latent or posture data (see Appendix D.3) (D) PCA projections of the posture time series (left) and latent state (right) color-coded according to bout category.

Here, the model's multivariate control impulse arose right before the onset of movement (which started at different time points in different trials), thus setting the initial conditions for the latent dynamical system (Figure 2B). After this point, the bout was then generated by quasi-autonomous dynamics, before decaying back to a fixed point. As shown in Figure 2A, the MGU could reconstruct bouts using a sparser control compared to the LDS.

Most of the information to generate a movement was contained in this low-dimensional impulse. Indeed, restricting the input to its initial peak was still sufficient to reconstruct bouts with $R^2 = 0.82$ across the test dataset. Although a single time-step impulse lasting for a single time bin (1.42ms) might appear unrealistically fast from a biological standpoint, our model was robust to changes in the duration of the control signal up to 10ms (Figure S2a). During this impulse signal, we found that, rather than using a one-hot encoding of the movement category, the different control units were continuously modulated in different ratios to generate specific bout types (see Figure S3). The state-space position at the onset of movement acts as a compressed representation of the forthcoming actions. Here, five principal components explained 95% of the variance at movement onset. This representation not only segregates known categories of movements but also retains continuous variations in kinematic features like frequency and amplitude (Figure S5). We found that the latent representation was robust across random initialization and splits of the training data, even those that excluded bout categories (see Figure S2B). Following this initial state, the movement unfolds by following the learned flow field. In the absence of an additional control signal, the trajectory in state space should therefore be untangled, with similar positions in state space leading to similar patterns in the near future (Russo et al., 2018). We tested this in the models, by measuring how well we could decode the category of movement from a single snapshot of the latent state (Figure 2C). In contrast to postural trajectories, which were highly tangled, the low-tangling of the latent state space allowed for high classification accuracy from that state. Surprisingly, the state space of the MGU 40ms after the control peak provided a higher classification accuracy even when compared with a linear classifier trained on the full time series of tail movements Figure 2C. This separation between

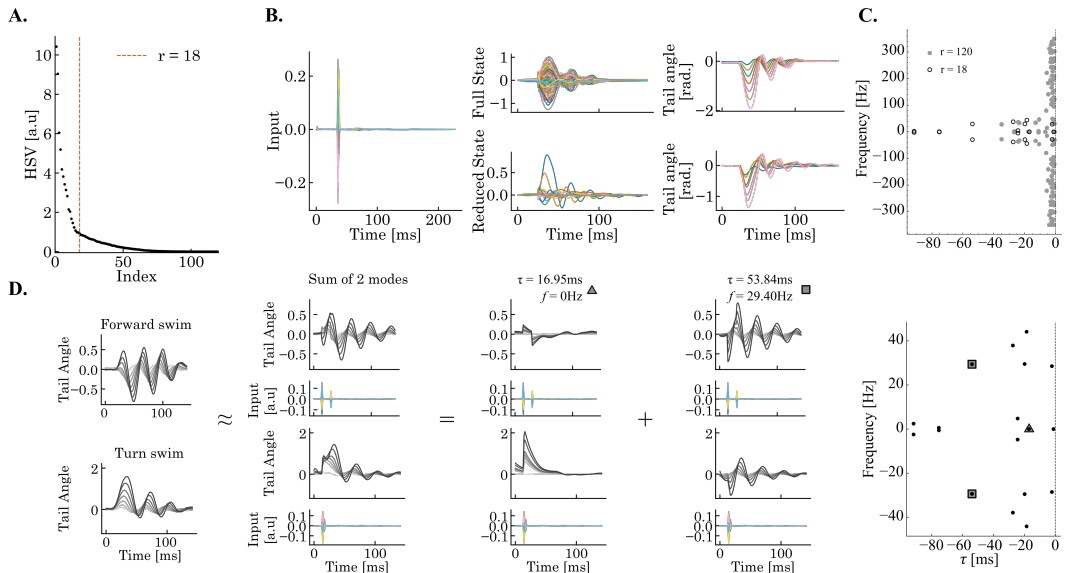

Figure 3: **Linear model reduction used to obtain compositional structures of behavioral repertoire.** (A) Hankel singular values of the system partitioned at $r = 18$ (red vertical line). (B) Input signal projected to full scale network $\boldsymbol{A}$ (top) and reduced network $\tilde{\boldsymbol{A}}^{(r)}$ (bottom) able to reconstruct swim of $R^2 = 0.818$. (C) Eigen spectrum of balanced truncated dynamics $\lambda(\tilde{\boldsymbol{A}}^{(r)})$ compared to the full state dynamics $\lambda(\tilde{\boldsymbol{A}})$. (D) Top two eigenmodes that capture the most variance are convolved through the input signal to construct forward swim (top) and turn swim with $R^2 = 0.76$ and $R^2 = 0.74$ , respectively. The oscillatory and decay mode are highlighted by a square and triangle in the truncated dynamics spectrum (right).

categories of movement cannot be attributed to a difference in the dimensionality of the latent space (see Appendix D.3). It demonstrates the quality of the representation of the movement within the latent space. Spreading of target variables in the neural state space has been shown to be associated with robustness to noise (Russo et al., 2018; Susman et al., 2021).

To benchmark our method, we compared it with the non-autonomous LFADS (Pandarinath et al., 2018) with inferred inputs and an autoregressive prior AR(1) for the control signal (see Appendix E). The method successfully reconstructed the postural observation ($R^2 = 0.94$). However, in this regime, the latent dynamical system was predominantly input-driven. Indeed, we could regress a projection of the control signal to display a correlation of r= 0.73 with the postural time series (see Figure S6). This leakage from the input data resulted in a less informative dynamical system, with a worse classification accuracy of the movement category from the latent space Figure 2C. This result suggests that for low-dimensional behavioral observation, sparse input priors are better suited at learning a meaningful dynamical system.

**Control of spatial navigation**   The control signal accounts for the influence of sensory and decision-making areas. As such, this signal is expected to encode information relative to navigational landmarks such as the relative position of a prey that the fish want to capture. To achieve its goal, the fish has to solve an inverse problem: determining what control signal will propel its body in a given position in space.

We propose that a straightforward mapping exists between the control impulse and the ensuing fish trajectory. This proposal is non-trivial, as our training dataset contained only posture information. Nonetheless, Figure S4 demonstrates that spatial displacement can be linearly predicted from the 120-dimensional initial latent state. The control signal and dynamic inferred by our method therefore allow for a simple sensorimotor coupling.

### 4.1.3 DISSECTING THE DYNAMICS

In the nonlinear dynamical system described above, input sparsity aids in dissecting the *control mechanisms*. Here, we transition to a linear dynamical system (LDS) to gain insights into the *dynamics*. Although LDS models did not match the performance of their nonlinear counterparts, their dynamics could be more easily interpreted. This illustrates a common tradeoff between the use of more expressive models – which can lead to more compressed representations, versus simpler models which can be more easily interpreted. We note that both types of models are accommodated by our framework, and can be switched between depending on the focus of the application.

Using iLQR-VAE, we fitted an LDS model using 120 latent dimensions and 10 control inputs. We truncated the system to the minimal dimension after which the decay in Hankel Singular Values started plateauing (see Figure 3). Here, this corresponded to 18 eigenvalues, and 10 distinct eigenmodes, corresponding to temporal decays of up to 100 ms and frequencies up to 45 Hz ( Figure 3c). This reduced model could accurately reproduce the output across a range of frequencies, given the sparse forcing inputs given to the original model (Figure 3B). Note that, as we show in Table S1, it was not possible to directly train models of such small size that were both sparse and yielded good reconstruction accuracy. In particular, when directly training linear models of smaller sizes, we were unable to find solutions that relied on sparse control inputs. Thus, model reduction was a necessary step to reach the smaller size models we needed to further dissect the dynamics. As we can see in Figure 3C, reducing the model corresponded to removing many eigenvalues corresponding to high frequencies and short time scales while adding eigenvalues with longer time scales and lower frequencies. The reduced system modes are therefore closer to the temporal and spectral components critical to generate the movement.

With our reduced model in hand, we were then able to decompose postural time series into the contributions of the individual modes. We focused on forward movements and turns, shown in Figure 3D (respectively top and bottom). Interestingly, turn movements were generated by the superposition of oscillatory (corresponding to complex conjugate eigenvalues) and non-oscillatory modes (with real-valued eigenvalue), while forward movements primarily relied on oscillatory modes (see Figure S7). This aligns with the findings from Huang et al. (2013), who identified populations of reticulospinal neurons essential for converting forward swims into turns. These neurons could be implicated in triggering the non-oscillatory modes we observe. This highlights how our model may facilitate the dissection of neural circuits responsible for generating movements, and help investigate how neural ablations or perturbations may relate to specific dynamic modes.

## 4.2 MODELING LOCOMOTION OF C. ELEGANS

To map the relevance of inferred control signals driving behavior requires controlled activation of a ground truth external stimulus. Here, we switched animal models, using a dataset from Broekmans et al. (2016) to study the input-driven dynamics of the C. elegans response to a 100ms aversive heat shock to the head. In response, C. elegans exhibited a robust escape sequence of behavioral maneuvers: reverse crawl, high-angle $\Omega$-turn to change heading direction, the forward crawl (Figure 4B). Unlike the larval zebrafish, the continuous locomotion of the C. elegans is captured as a mixture of 4 principal modes, termed eigen worms (EW), to capture its 2D body posture ( Figure 4A).

We trained a LDS of network size $n = 60$ driven by $m = 5$ inputs which could accurately recover the continuous dynamics (Figure 4C). Across all 90 trials, a salient input peak was aligned on the heat shock activating the reverse crawl (Figure 4D). Moreover, there is a secondary peak driving the $\Omega$-turn changing the heading direction away from the stimuli. While driving inputs were less sparse due to the continuous locomotive nature of the C. elegans locomotion, large inputs were recruited in the transition between behavioral maneuvers.

The choice of LDS model allowed for balanced model truncation, (Figure S8), where we interpreted how the task-related inputs influenced the dynamics of behavioral switches in the escape sequence. We projected the three slowest decaying eigenmodes of the reduced model into the EW space. Each mode contribution reflected specific behavioral maneuvers, shown in Figure S8C. The frequency of modes that were recruited in forward and reversal crawls aligned with results found in Costa et al. (2023). The reduced LDS provides an alternative approach to segment behavioral states in continuous locomotion and infer switches between behavioral modes.

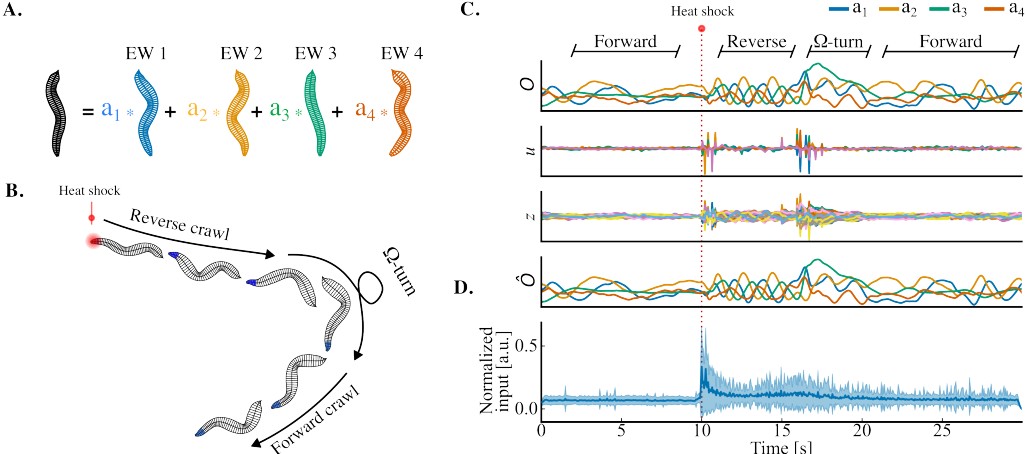

Figure 4: **iLQR-VAE captures C. elegans continuous locomotion and infers heat shock-induced behavioral transitions.** (A) C. elegans 2D body posture was expressed as a combination of 4 eigen-worm (EW) postures (95% explained variance). (B) Time-lapse of C. elegans body posture following an aversive heat shock that triggers a behavioral sequence of reverse crawling, high angled $\Omega$-turn, and forward crawling away from the stimuli. (C) (From top to bottom) example trial of the postures around the heat shock (red marker). The inferred driving inputs align with the heat shock at $t = 10$ s, with a secondary burst of inputs transitioning the reversal mode to a $\Omega$-turn. The reconstruction of the posture from the latent dynamics $\hat{O}$ matches the observation $O$. (D) Mean $\pm$ standard deviation across all escape trials of the normalized L2-norm of the control is aligned to the external heat shock.

## 5 DISCUSSION

In this work, we introduced a novel framework to learn the dynamics underlying the generation of movement from unstructured postural datasets. Our approach simultaneously identifies the control signals that drive actions and the dynamical systems that generate them. Our empirical findings revealed that temporally sparse control signals are sufficient to replicate the complete locomotor repertoire of larval zebrafish. In a dataset of C. elegans continuous crawling recordings, our model could infer control signals that were aligned with external perturbations. While high-dimensional expressive dynamics were necessary to model the data accurately, we proposed to use balanced truncation to extract a minimal, interpretable linear dynamical system model of the data. In both modelled species, our approach yielded insights consistent with prior findings.

While balanced truncation is a well-established method in the control literature, it had not yet been combined with data-driven approaches to learning behavioral dynamics. Indeed, to apply balanced reduction, one must have access to all the parameters of a controlled dynamical system. If the inputs to the system are unobserved (as is often the case in behavioral datasets), it is thus necessary to take the approach we did, and propose constraints on the inputs that allow to infer the most likely inputs to have generated the data.

A limitation of our work is that we relied on two separate dynamical systems to be able to extract interpretable inputs and interpretable dynamics from the data. This could be handled in a more unified manner in the future, for instance by obtaining the LDS by linearization (Sussillo & Barak, 2013), or by co-training the linear and MGU models, as in Smith et al. (2021).

Overall, separating out the contributions of intrinsic dynamics and control signals that shape animal movements can be helpful for interpreting neural recordings, and provide insight into how brains control movement. Our model could be used to estimate how either control signals, or output dynamics, change in different behavioral contexts, or pathological states, and make mechanistic predictions testable with brain imaging studies. In subsequent research, our framework could be adapted to other animal behavior, such as limb-based locomotion in flies or mice. Alternative control priors could also be investigated as an alternative to temporal sparsity.

ACKNOWLEDGMENTS

We would like to thank Il Memming Park for insightful discussions on the manuscript. T.S.M received funding from the European Union's Horizon 2020 research and innovation program under the Marie Skłodowska-Curie grant agreement $813457$. C.K.M. acknowledges support from the Simons Collaboration on the Global Brain (543009); C.K.M and A.J. received support from Fundação para a Ciência e a Tecnologia (FCT-PTDC/BIA-OUT/32077/2017-IC&DT-LISBOA-01-0145-FEDER) M.B.O received support from the European Research Council (ERC NEUROFISH 773012) and Volkswagen Stiftung "Life?" initiative. A.J was supported by a Fellowship from the La Caixa Foundation (LCF/BQ/PR20/11770007). M.S. was funded by EPSRC DTP studentship. This work was performed using resources provided by the Cambridge Service for Data Driven Discovery (CSD3) operated by the University of Cambridge Research Computing Service (www.csd3.cam.ac.uk), provided by Dell EMC and Intel using Tier-2 funding from the Engineering and Physical Sciences Research Council (capital grant EP/T022159/1), and DiRAC funding from the Science and Technology Facilities Council (www.dirac.ac.uk). We thank Pedro Tomás Martins Silva and Alexandre Laborde for their help in collecting the zebrafish dataset. Fish husbandry was supported by the Champalimaud Foundation Fish Facility, which received support from the research infrastructure CONGENTO, co-financed by the Lisboa Regional Operational Programme (Lisboa2020), under the PORTUGAL 2020 Partnership Agreement, through the European Regional Development Fund (ERDF), and the Fundação para a Ciência e Tecnologia (Portugal) under the project LISBOA-01-0145-FEDER-022170.

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

# Appendix

## A  SUMMARY OF THE ILQR-VAE METHOD

iLQR-VAE is a recently proposed method to learn input-driven dynamical systems from data. In this section, we provide a brief description of the various components of the methods. More details can be found in Schimel et al. (2022). iLQR-VAE models time series of observations $\boldsymbol{o}$ as arising from an input-driven nonlinear latent dynamical system. The generative models considering in iLQR-VAE have the following form :

$$\text{latent state} \quad \boldsymbol{z}_{t+1} = f_\theta(\boldsymbol{z}_t, \boldsymbol{u}_t, t) \tag{S1}$$
$$\text{observations} \quad \boldsymbol{o}_t|\boldsymbol{z}_t \sim p_\theta(\boldsymbol{o}_t|\boldsymbol{z}_t), \tag{S2}$$

where $\boldsymbol{z}$ denote latent trajectories, $\boldsymbol{u}$ the inputs driving those trajectories, and $\theta$ are model parameters. Inputs in iLQR-VAE are used to capture process noise, to set the initial condition of the dynamics, and to model control inputs driving the dynamics. The parameters $\theta$ are learned by optimizing the evidence lower-bound (ELBO) (Kingma & Welling, 2013), a lower bound on the log-likelihood of the observations given the model parameters, $p_\theta(\boldsymbol{o})$. Computing the ELBO requires introducing an approximate posterior distribution $q(\boldsymbol{u}|\boldsymbol{o})$. The ELBO objective then takes the following form :

$$\mathcal{L}(\mathcal{O}, \theta, \phi) = \sum_k \mathbb{E}_{q_\phi(\boldsymbol{u}|\boldsymbol{o}^{(k)})} \left[ \sum_{t=1}^{T} \log p_\theta(\boldsymbol{o}_t^{(k)}|\boldsymbol{z}_t) + \log p_\theta(\boldsymbol{u}_t) - \log q(\boldsymbol{u}_t|\boldsymbol{o}^{(k)}) \right] \tag{S3}$$
$$\leq \log p_\theta(\mathcal{O}). \tag{S4}$$

with respect to both $\theta$ and $\phi$. In iLQR-VAE, the approximate posterior over inputs for a set of observations $\boldsymbol{o}^{(k)}$ is defined as a Gaussian distribution, centered at the maximum of the true posterior over inputs,

$$\boldsymbol{u}^\star(\boldsymbol{o}^{(k)}) = \underset{\boldsymbol{u}}{\operatorname{argmax}} \ \log p_\theta(\boldsymbol{u}|\boldsymbol{o}^{(k)}). \tag{S5}$$

This set of inputs can be obtained as the solution of an optimization problem, which is solved here using the iLQR algorithm (Li & Todorov, 2004). iLQR-VAE then uses a shared set parameters to model the posterior covariance. Those consist of a set of spatial covariance parameters $\boldsymbol{\Sigma}_\text{s}$, and a set of temporal covariance parameters $\boldsymbol{\Sigma}_\text{t}$). The full covariance is then defined as the Kronecker product of those terms.

Given this parameterization of the approximate posterior, iLQR-VAE then optimizes the ELBO by drawing samples from $q(\underline{\mathbf{u}}|\underline{\mathbf{o}}^{(k)})$ and using the reparameterization trick (Kingma et al., 2015) to obtain gradients.

## B  DETAILS OF THE GENERATIVE MODEL

### B.1  PRIOR DISTRIBUTION

We used a Student-t distribution as our prior over inputs. This took the following form :

$$p_\theta(\mathbf{u}_t) = \frac{\Gamma\left[(\nu+m)/2\right]}{\Gamma\left[\nu/2\right](\nu\pi)^{m/2}|\mathbf{S}|} \left[ 1 + \frac{1}{\nu} \mathbf{u}_t^T \mathbf{S}^{-2} \mathbf{u}_t \right]^{-(\nu+m)/2}. \tag{S6}$$

where $\mathbf{S} = \operatorname{diag}(s_1, \ldots, s_m)$. The Student-t prior in Equation (S6) arises from the following construction; at every point in time, the scale of the input variance $\sigma_u$ is sampled from $\chi^{-1}$ distribution with $\nu$ degrees of freedom, i.e $\sigma_u \sim \chi^{-1}(\nu)$ – where larger values of $\nu$ shift the distribution towards a larger mean. In a second step, inputs are sampled from a Gaussian distribution whose variance $\boldsymbol{S}$ is scaled by $\sigma_u$, i.e $\boldsymbol{u} \sim \mathcal{N}(0, \sigma_u \boldsymbol{S})$. The advantage of this construction is that it promotes the use of small inputs most of the time, whilst still accommodating the occasional large input.

## B.2 DYNAMICS

To fit the larval zebrafish data, we selected a Minimal Gated Unit (MGU) RNN presented by Heck & Salem (2017). The dynamics are defined as,

$$\boldsymbol{f}_t = \sigma(\boldsymbol{U}_f \boldsymbol{z}_{t-1}) \tag{S7}$$

$$\hat{\boldsymbol{z}}_t = \phi(\boldsymbol{U}_h(\boldsymbol{f}_t) \odot \boldsymbol{z}_{t-1}) + \boldsymbol{W}\boldsymbol{u}_t \tag{S8}$$

$$\boldsymbol{z}_t = (1 - \boldsymbol{f}_t) \odot \boldsymbol{z}_{t-1} + \boldsymbol{f}_t \odot \hat{\boldsymbol{z}}_{t-1} \tag{S9}$$

where the non-linearity functions $\sigma$ and $\phi$ were chosen to be $\tanh$ functions.

## C DETAILS OF THE PENDULUM EXAMPLE

We generated observations from a synthetic system consisting of two driven pendulums. The equations describing the autonomous dynamics of the system are as follows :

$$m\ddot{\theta}_1 = -g/L \sin\theta_1 + k\theta_2 - f\dot{\theta}_1 \tag{S10}$$

$$\approx -g/L\theta_1 + k\theta_2 - f\dot{\theta}_1 \tag{S11}$$

$$m\ddot{\theta}_2 = -g/L \sin\theta_2 + k\theta_1 - f\dot{\theta}_2 \tag{S12}$$

$$\approx -g/L\theta_2 + k\theta_1 - f\dot{\theta}_2, \tag{S13}$$

where $\theta_1$ and $\theta_2$ are the angles of the pendulums relative to their equilibrium position. In Equation (S10), $m = 1\text{kg}$ is the pendulum mass, $g = 9.8\text{ms}^{-2}$ is the gravitational acceleration term, $L = 1\text{m}$ the length of the pendulum, $k = 160\text{Nm}^{-1}$ the strength of the coupling spring in-between the pendulums, and f the damping term. As the angles of the pendulums remained small, we approximated the dynamics as linear.

We modeled the effect of inputs forcing switches in the pendulum motion by integrating the system's position from random initial conditions, and stitching together several trajectories. In both our training and test sets, the dynamics were driven by either 1 or 2 inputs. We sampled observations at time intervals of $dt = 0.06\text{s}$. Note that even though we only had two pendulums, the differential equation underlying the dynamics is of order two, such that modeling the dynamics of this system requires a 4-dimensional linear dynamical system.

To recover the coupling constant from the frequencies learned by the model, we used the fact that the two oscillatory modes have frequencies of $\omega_0$ and $\omega_1 = \sqrt{\omega_0^2 + 2k}$, where $\omega_0 = g/L$. As such, $k = (\omega_1^2 - \omega_0^2)/2$.

### C.1 RECOVERING THE DYNAMICS OF DRIVEN COUPLED PENDULUMS

To illustrate the use of our approach, we first validated its ability to recover the true dynamics of a system on a synthetic example. We generated data from a toy dynamical system consisting of a set of two lightly damped pendulums, coupled by a spring, and driven by sparse forcing inputs (Figure S1). We modeled the pendulums as linear systems, as they remained in the small angle regime throughout the experiment. The state trajectories of these coupled pendulums (which were 4-dimensional) were then projected onto a lower-dimensional (2D) space and corrupted with Gaussian noise to give rise to a set of observations. The equations governing the system's motion are described in Equation (S10).

We fitted the observations using a 20-dimensional latent linear dynamical system. Computing the balanced realization of this model revealed the existence of 4 dominant modes in the dynamics (see Figure S1B). Reducing the dynamics to 4 dimensions thus allowed us to recover 4 dominant eigenvalues, corresponding to two oscillatory modes (Figure S1C). From these eigenvalues, we were able to recover the coupling constant of the underlying synthetic system highly accurately (see details in Appendix C), with the true coupling constant $k = 160 \text{ Nm}^{-1}$ falling within the error of our prediction, $k_{pred} = (160 \pm 1)\text{Nm}^{-1}$ across 4 random initializations and data splits. This highlights how combining iLQR-VAE with balanced model reduction allows to recover the minimal linear dynamical system that can describe the data, and to extract valuable information about the underlying dynamical system.

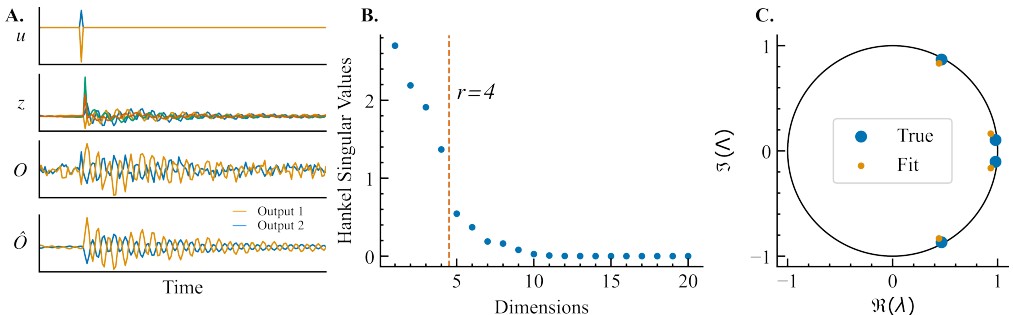

Figure S1: **System Identification** (A) Latent coupled oscillator driven by unknown sparse forcing inputs. The control signal $u$ is fed to coupled pendulums parameterized by $z$. iLQR-VAE learns from observations $O$ and can reconstruct $\hat{O}$ after training. Here, blue and orange are used to denote the two input channels of the model, and the two observed outputs of the model. (B) Spectrum of the Hankel singular values. (C) Ground truth eigenvalues of the dynamics (blue), and eigenvalues of the truncated model (orange).

## D MODEL COMPARISON

### D.1 MODEL SELECTION

The linear dynamical system and MGU RNN were benchmarked on the zebrafish dataset. We evaluated each model's performance in reconstructing the tail dynamics and determining the sparsity of the control signal required to generate the dynamics, see Table S1. Across all combinations of network types and input dimensions, all models achieved high $R^2$ scores in the validation dataset. However, the sparsity of the control signal varied, depending on the scale of the network and input dimensions throughout the parameters space.

Table S1: Benchmark model parameters (mean $\pm$ SEM)

| n | m | MGU | Linear | $\mathbf{R}^2$ | $\|\|u\|\|_1/\|\|u\|\|_\infty$ |
|---|---|-----|--------|-------|-------------------|
| 60 | 5 | ✓ | - | 0.974 | 21.7 |
| 60 | 10 | ✓ | - | 0.975 | 13.9 |
| 90 | 5 | ✓ | - | 0.976 | 7.88 |
| 90 | 10 | ✓ | - | 0.976 | 9.43 |
| 90 | 15 | ✓ | - | 0.976 | 4.59 |
| 120 | 5 | ✓ | - | 0.976 | 8.26 |
| 120 | 10 | ✓ | - | $0.983 \pm 0.000$ | $2.29 \pm 0.05$ |
| 120 | 15 | ✓ | - | $0.986 \pm 0.000$ | $4.71 \pm 0.03$ |
| 120 | 20 | ✓ | - | 0.974 | 3.78 |
| 120 | 10 | - | ✓ | $0.971 \pm 0.000$ | $2.75 \pm 0.03$ |
| 80 | 10 | - | ✓ | $0.979 \pm 0.00$ | $3.06 \pm 0.08$ |
| 60 | 10 | - | ✓ | $0.985 \pm 0.00$ | $3.97 \pm 0.10$ |
| 40 | 10 | - | ✓ | $0.986 \pm 0.00$ | $5.76 \pm 0.20$ |
| 20 | 10 | - | ✓ | 0.987 | 15.84 |
| 18 | 10 | - | ✓ | $0.985 \pm 0.002$ | $23.5 \pm 6.2$ |
| 15 | 10 | - | ✓ | $0.985 \pm 0.000$ | $28.7 \pm 5.2$ |

## D.2 Model robustness

We evaluated the robustness of the model fit across random initializations. To do so, we apply Singular Vector Canonical Correlation Analysis (SVCCA) to the Principal Component Analysis (PCA) projection of the initial condition of the RNNs, at movement onset Raghu et al. (2017). SVCCA allows to compare representations by measuring their similarity in a way that is independent to rotations of the data. SVCCA is based on Canonical Correlation Analysis (CCA), but includes an additional step during which the dimensionality of the whole data is reduced, greatly speeding up the analysis. In Figure S2, we applied SVCCA to a dataset of size $K \times P$ where $P$ was the number of points in the dataset, and $K$ the dimension of the PCA projection of the initial condition.

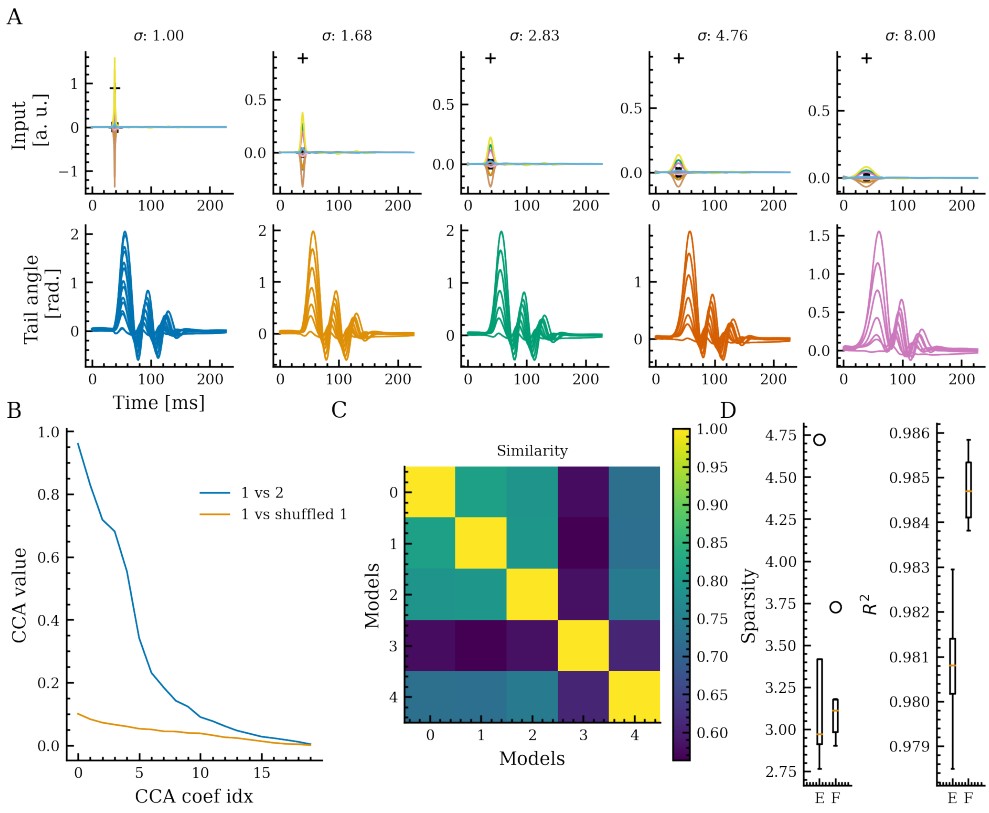

Figure S2: **Robustness of the learned model.**(a) Smoothing the input signal to generate movements using a Gaussian kernel $\sigma = \{1.00, 1.68, 2.83, 4.76, 8.00\}$ had small effect on reconstructing the original movement. (b) Using SVCCA to compare the initial latent state representation of two independently initialized models (blue) and a control with shuffled labels for the initial state of the same model (orange). (c) The similarity matrix of the top 3 principal components of the initial state across 5 independently initialized models. (d) A comparison of sparsity and reconstruction for these models when trained on the full bout dataset (F) and the dataset excluding a specific swim type during training (E).

## D.3 Classification of movement category from the latent space

Here we detail the method to assess the ability of the latent representation to untangle movement from different categories. At every time step we trained and tested a linear classifier using a 80%/20% split of the latent state trajectory at that time. The $n = 120$ dimensional latent state of the MGU and LDS were projected into their respective 20 first principal components as a regularization. We only considered movement presenting a positive first half beat so that rectification would not be required to identify the movement category. With 11 categories and a balanced dataset the chance level was at 9%.

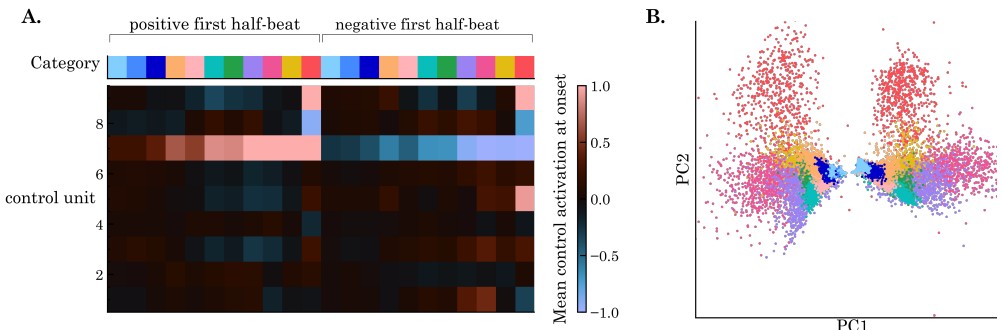

Figure S3: **MGU control peak for different category of movement**. (A) Average activation during the first peak for each category of tail movement (B) 2D projection of the 10-dimensional peak in control signal. The control continuously parameterize the movement category

## D.4 LINEAR PREDICTION OF TRAJECTORY FOLLOWING INITIAL CONTROL IMPULSE

We measured if the initial control input used to drive the network was predictive of navigational landmarks. By aligning input signals by their initial peak and defining the initial state of the network, we fit a linear regression of the initial 120-dimensional state of the network to each navigation feature, Figure S4. This was further constrained by $L1$-Norm regularization.

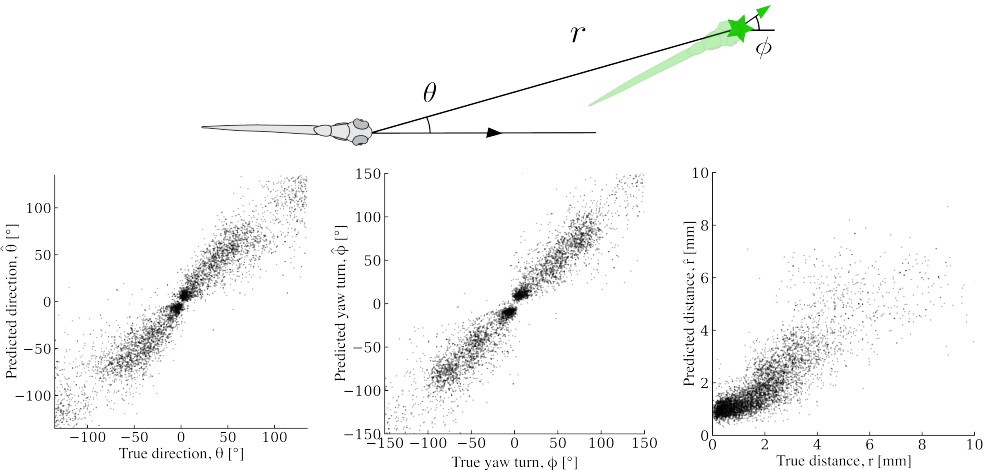

Figure S4: **Linear decoding of navigation features following the first control peak of the network**. Schematics demonstrating important movement features to navigate in the environment are swim direction $\theta$, swim yaw $\phi$, and swim distance $r$. Linear regression with L1 constraint of the initial network state (state reached after the peak control signal) to predict swim trajectory direction ($R^2 = 0.90$), turn yaw ($R^2 = 0.93$), and swim distance ($R^2 = 0.72$).

## D.5 LINEAR PREDICTION OF TAIL MOVEMENT FEATURES

We extracted the tail frequency and the maximum tail amplitude from the dataset. Using a similar method to Appendix D.4, we could predict the tail beat frequency ($R^2 = 0.85$) and maximum tail bend amplitude ($R^2 = 0.96$).

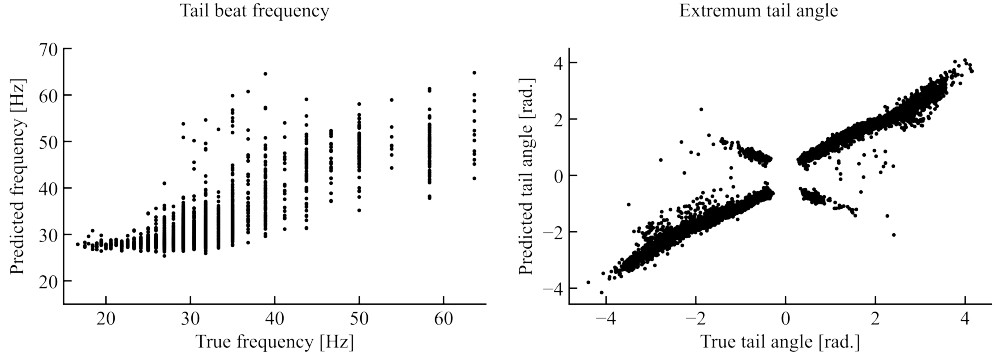

Figure S5: **Linear decoding of tail oscillations features following the first control peak of the network**. Linear regression with $L1$-Norm constraint of the initial network state (state reached after the peak control signal) to predict the tail beat frequency ($R^2 = 0.85$) and maximum tail beat amplitude ($R^2 = 0.96$).

## E  IMPLEMENTATION OF LFADS

The non-autonomous LFADS model with inferred inputs is composed of RNNs in the encoder, controller and generator. The encoder is constituted of bidirectional RNNs. The controller RNN generates the control signal from the encoder and past factors. To facilitate comparison with iLQR-VAE, we used a similar architecture where the generator RNN hidden state is directly read out into the output. As in Pandarinath et al. (2018), we used an autoregressive prior AR(1) for the control signal. During training, we implemented a linear KL warmup from 0 to 1 during the first 100 epochs. See Table S2 for important hyperparameter settings.

Table S2: LFADS Hyperparameters

| | |
|---|---|
| RNN type | GRU |
| Encoder Dimension | 64 |
| Controller Dimension | 32 |
| Control Dimension | 10 |
| Generator Dimension | 120 |
| AR(1) tau | 10 |
| AR(1) noise variance | 0.1 |
| Dropout rate | 0.3 |
| Batch size | 75 |
| Learning rate | 3e-4 |

## F  IMPLEMENTATION OF BALANCED MODEL REDUCTION

### F.1  STATE CONTROLLABILITY AND OBSERVABILITY

The controllability of the LDS is determined by, both, the coupling of the state dynamics $\boldsymbol{A}$, and, the number of control inputs in the LDS $m$. The controllability matrix $\mathcal{C}$ measures the behavior in which the input matrix propagates through the state dynamics $\boldsymbol{A}$, defined as,

$$\mathcal{C}(\mathbf{A}\ \mathbf{B}) = \begin{bmatrix} \mathbf{B} & \mathbf{AB} & \dots & \mathbf{A}^{n-1}\mathbf{B} \end{bmatrix} \tag{S14}$$

where $\mathcal{C} \in \mathbb{R}^{n \times (m*n)}$. If the control can reach all directions in the $\boldsymbol{A}$ subspace, the LDS is completely controllable, thus the $\mathcal{C}$ will have column rank $n$. In such case, any state in the network from an initial state can eventually be reached by an arbitrary input signal $\{\boldsymbol{u}\}$.

In contrast, the observability matrix $\mathcal{O}$ assesses whether all states within the LDS can be realized from its measured observable $\boldsymbol{o}$. $\mathcal{O}$ measures how much information can be retrieved from the

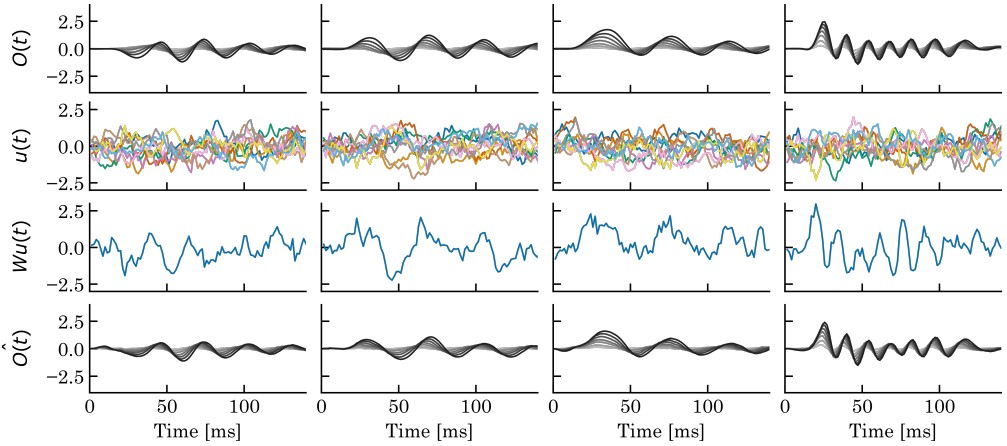

Figure S6: **Example of LFADS AR(1)** Each columns corresponds to an example movements. By regressing the 10-dimensional control signal to the last tail segments across the dataset we found a projection of the input $Wu(t)$ presenting a correlation of r=0.73 with the last tail segments of the input movement O(t).

readout matrix $C$ about the state dynamics $A$, defined by,

$$\mathcal{O}(\mathbf{C}; \mathbf{A}) = \begin{bmatrix} \mathbf{C} \\ \mathbf{CA} \\ \vdots \\ \mathbf{CA}^{n-1} \end{bmatrix} \tag{S15}$$

where $\mathcal{O} \in \mathbb{R}^{(n_o * n) \times n}$. The LDS is observable providing $\mathcal{O}$ has full rank $n$, otherwise regions in the state space cannot be reached via observations $o$. Observability is important in order to control the LDS.

## F.2 IDENTIFICATION OF BALANCED MODEL TRANSFORMATION

In order to find balanced basis transform, $T$ for the LDS, we employed the framework from Laub et al. (1987). Due to the LDS state dynamics being constrained to be asympotically stable, both, $W_c$ and $W_o$ are positive definite. Therefore, we can compute the Cholesky factors of the Gramians,

$$\mathbf{W}_c = \mathbf{L}_c \mathbf{L}_c^T, \tag{S16}$$

$$\mathbf{W}_o = \mathbf{L}_o \mathbf{L}_o^T \tag{S17}$$

where $L_c$ and $L_o$ are the lower triangles of the respective factorization. Using the product of the two Cholesky factors, $L_c^T L_o$, we computed the singular value composition,

$$\mathbf{L}_c^T \mathbf{L}_o = \mathbf{U} \mathbf{\Lambda} \mathbf{V}^* \tag{S18}$$

where $U \in \mathbb{R}^{n \times n}$ and $V^* \in \mathbb{R}^{n \times n}$ are right and left basis, respectively, and $\Lambda \in \mathbb{R}^{n \times n}$ is a diagonal matrix where $\Lambda = \Sigma^2$ corresponding to the Hankel singular values. From this decomposition, the balancing transformation can be obtained, yielding,

$$\mathbf{T} = \mathbf{L}_c \mathbf{V} \mathbf{\Lambda}^{\frac{1}{2}} \tag{S19}$$

$$\mathbf{T}^{-1} = \mathbf{\Lambda}^{-\frac{1}{2}} \mathbf{U}^* \mathbf{L}_o^T. \tag{S20}$$

This basis is then projected on the LDS, yielding a system to that of Equation (7).

## F.3 COMPARISON OF REDUCED MODELS

We varied the order of truncation, $r$, and compared the balanced truncated model performance in reconstructing the data. We found that the LDS truncated to $r = 18$ could reconstruct most movements, with the exception of escape swims.

Table S3: Reduced models reconstruction

| r | $R^2$ |
|---|---|
| 120 | 0.970 |
| 60 | 0.960 |
| 40 | 0.904 |
| 20 | 0.770 |
| 10 | 0.119 |

### F.4 CONTRIBUTION OF EIGEN MODES OF BALANCED TRUNCATED TO TURNS AND FORWARD MOVEMENTS

Using our LDS truncated to r=18, we can decompose the readout of the dynamical system as the sum of the contribution of each mode. When we quantify the contribution of each mode to the final tail movement we found that overall routine turns are most differentiated from slow2 forward movement in their use of the mode corresponding a real eigenvalue with $\tau$=-17ms similar to the examples in Figure 3D.

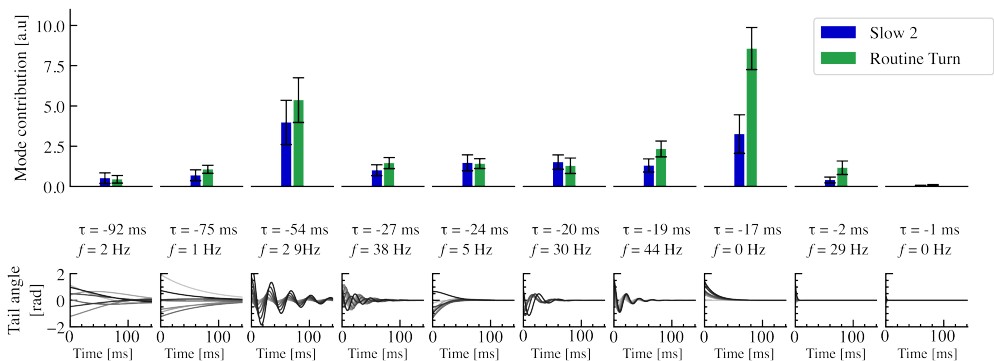

Figure S7: **(Top)** Distribution of the $L^2$-norm of each mode across routine turn and slow2 movements. **(Bottom)** Projection of impulse response into the readout for each of the 10 modes (8 complex conjugates eigenvalues and 2 real eigenvalues).

## G ZEBRAFISH DATASET

**Fish Care** Adult fish were maintained at 25°C on a 14:10 hour light cycle following standard methods. Embryos were collected and larvae were raised at 28°C in E3 embryo medium (5 mM NaCl, 0.17 mM KCl, 0.33 mM CaCl2 and 0.33 mM MgSO4) in groups of 30. The wild-type line Tübingen (Tu) was used. Sexual differentiation occurs at a later larval stage, therefore the sex of the animals cannot be reported. All experimental procedures were approved by the Champalimaud Foundation Ethics Committee and the Portuguese Direcção Geral Veterinária, and were performed according to the European Directive 2010/63/EU.

**Behavioral experiments** The dataset was compiled from observations from over 100 freely-swimming larvae at 6 or 7 days post fertilization. The larvae were exposed to various ethologically relevant visual and auditory stimuli, including moving gratings of different speeds and orientations, and transitions between light and dark environments. We monitored the larvae's movements by tracking seven landmarks along their tails The oscillations of the tail, driven by activity in the spinal cord, were characterized by measuring the angle of each segment relative to the fish trunk. Using methods from Marques et al. (2018), we segmented the data into swim bouts which we classified into 11 distinct kinematic categories: approach swim, slow 1, slow 2, burst swim, J turn, high angle turn, routine turn, shadow avoidance, O bend, long latency C start, short latency C start (see Figure 1C and Supplementary Video 1). We excluded capture swim categories since the dataset did not

contain prey capture episodes. For each category, the dataset contained both movements presenting a positive first half beat as in Figure 1C or a negative first half beat. The final dataset was balanced and contained 30800 tail angle time series lasting 230ms.

# H    Details of C. elegans

## H.1    C. elegans dataset

**Behavioral experiments**    We used the dataset from Broekmans et al. (2016), composed of two experiments. First, a foraging dataset where 12 worms were recorded at 16 Hz for 35 min. The other set consisted of 98 worms recorded at 20 Hz for 30 s. Each worm was subjected to a 100 ms shock on the head with an infrared laser 10 seconds after acquisition and tracked for an additional 20 s post-zap. The frame rate was down sampled to 16Hz, and recordings were segmented into 12.5 s trials. To train the LDS model we used these 12.5 s trials. To validate reconstruction of the dynamics and inferred inputs, we use the entire 30 s trials for each worm in the escape dataset.

**Behavioral tracking**    C. elegans behavior was acquired by tracking 100 segments along the body. At each segment, the tangential angles along the centerline was calculated. Four principal components explain over 95% of the variation, and each mode was referred to as an eigen worm (EW), shown in Figure 4. EW1 and EW2 map as the sine-cosine pair of the body posture, different combinations reflect a wave along the body at different phases, important for forward and reverse crawl maneuvers. EW3 represents the near-constant curvature along the central half of the body line, this mode is mainly recruited for turns. Finally, EW4 denotes the bend in the head and tail of the posture.

## H.2    Reduced model eigen modes capture behavioral states in escape sequences

We applied balanced model reduction to enhance our understanding of the LDS model's dynamics. In Figure S8A, we compressed the LDS model, size $n = 60$, to $r = 20$ with balanced model truncation. The reduced model was able to reconstruct the EW dynamics with $R^2 = 0.702$. The model with 20 states covered a wide spectrum of frequencies and timescales comparable to those in the full LDS model, as shown in the eigen spectrum Figure S8B. Interestingly, the three slowest decaying modes mapped to the reverse, $\Omega$-turn, and forward crawl, see Figure S8C.

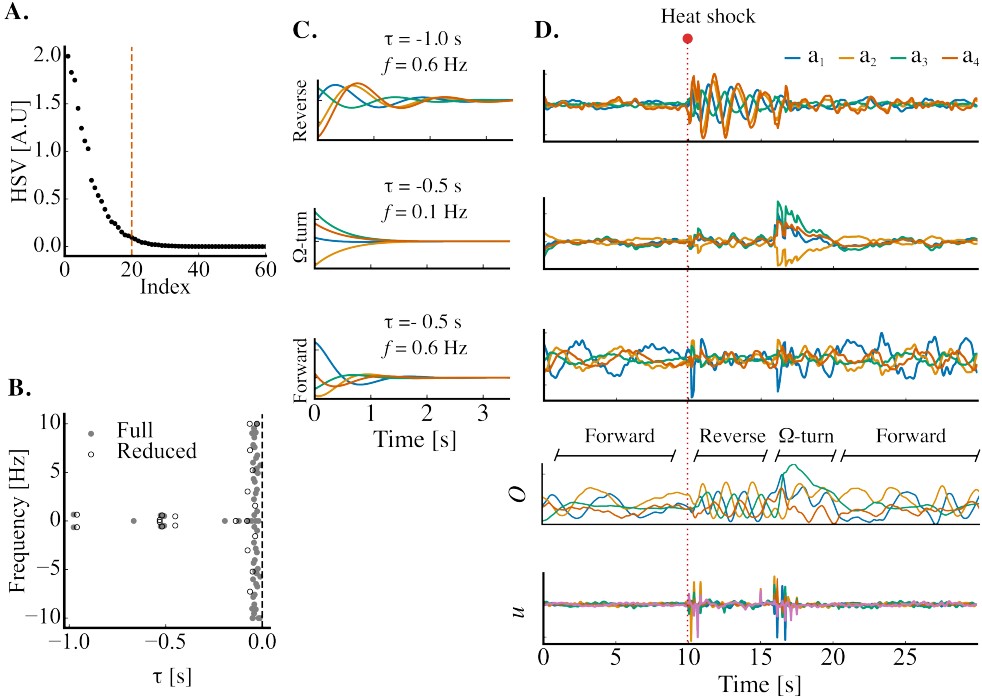

Figure S8: **Eigen-modes from balanced model reduction reflect behavioural states in escape sequence.** (A) Hankel singular values of the LDS truncated at $r = 20$ (red vertical line). (B) Eigen spectrum of the reduced model (open marker) compared to the full LDS (black marker). (C) From top to bottom, are the projections of an impulse response for the three slowest decaying eigen modes in the reduced model. (D) The individual contribution of the three slowest eigen modes capture the highest variance across at different behavioral states within the escape sequence. Moving from top to bottom, each modes corresponds to the reverse crawl, escape $\Omega$-turn, and forward crawl, respectively. Their contribution is reflected in the in the full reconstruction (middle-bottom panel). The driving-input to generate mode contribution (bottom).

