# OpenReview forum: "Learning interpretable control inputs and dynamics underlying animal locomotion"
_ICLR.cc/2024/Conference — ICLR 2024 poster_

### Official Review · Reviewer_j5DC · 2023-10-31

**Soundness:** 4 excellent
**Presentation:** 4 excellent
**Contribution:** 3 good
**Rating:** 6
**Confidence:** 3

**Summary:**

This manuscript provides a framework for modeling the behavior of the larval zebrafish using a control theoretic and dynamical systems framework. They model the behavior of a system as arising from a dynamical system driven by sparse inputs with a students-t distributed prior. They assume behavior is generated from the dynamical system by either a linear or RNN model, and use iQLR-VAE to infer model parameters and the sparse inputs. In each case they use a model truncation approach to reduce the number of dimensions considered. They begin by validating the approach on a double pendulum oscillator system, showing they could reproduce the number and values of eigenvalues. Then they analyze a dataset of larval zebrafish behavior, which is organized into discrete bouts and thus a natural test case for this approach. They find they need high capacity models (120 latent dimensions, 10 control inputs) to fit the behavior. They show that the resulting representation is useful, in that the dynamical representations provides a more disentangled input that can be better decoded from the top 5 PCs compared to the postural representation or even the full time series in some cases. By using model reduction, they show the dynamics underlying behavior can be decomposed  into separate modes.

**Strengths:**

* The formulation is well described and I believe novel, and an addition to other methods fitting dynamical systems to neural and muscular systems, but here in the case of sparsely driven behavior.

* The result about better disentangling of behavior is interesting, and the spatial decoding result as well.

* The model reduction approach to achieve small models is interesting.

**Weaknesses:**

* There is a lack of benchmarks comparisons to other techniques in the literature such as LFADs. Even though some suggest continuous inputs for their control scheme, could they be used to model the behavioral dynamics in Figure 3C for instance.

* I found the applications a bit limited. It wasn’t obvious from this analysis that there were fundamentally new types of experiments that were enabled by this approach.

* Overall, I think the lack of benchmarks combined with the lack of novel applications was a major sticking point for me and I view this manuscript as very borderline. I would be happy to reconsider the manuscript (which was very well written) if these were added or addressed. I see this as just 1 compelling figure short of a reasonable submission.

**Questions:**

* Does the number of control inputs match the number of bout types people have reported in the literature?

* There is no ground truth in behavior for the driving inputs and thus there is no way to validate that they are correct, however with an appropriate experiment (e.g. applying a perturbation or stimulus at a specific time) perhaps they could be useful.

* Generally, more concretely linking any of the observations to real neural or behavioral variables would improve the manuscript's impact.

---

> ### Author Response · Authors · 2023-11-20
> **Response to j5DC**
>
> Many thanks for your in-depth review and for the positive feedback! As described in the common reply, we have now updated the paper to reflect new results, which we believe should address the majority of your concerns. Below, we address the specific points/questions you raised. Of course, please let us know if anything is unclear or if you feel that we did not properly address your concerns.
>
> > There is a lack of benchmarks comparisons to other techniques in the literature such as LFADs. Even though some suggest continuous inputs for their control scheme, could they be used to model the behavioral dynamics in Figure 3C for instance.
>
> That is a good point. We have now included a comparison to LFADS (where we used the standard LFADS implementation, with an autoregressive prior). In summary, we found that LFADS learns inputs throughout the whole time series that exhibit a strong correlation with the actual behavior (see **new Figure S6**). This is somewhat expected, as the autoregressive prior does not discourage the model from using inputs to fit the temporal structure in the data, that could alternatively be captured by the dynamics. To address the second part of your question, we found that such inputs allow us to accurately reconstruct the behavior, but lead to a representation in the LFADS model that is less predictive of the movement category than the iLQR-VAE MGU representation (see **updated Figure 2.C**).
>
>
> > I found the applications a bit limited. It wasn’t obvious from this analysis that there were fundamentally new types of experiments that were enabled by this approach.
>
> > There is no ground truth in behavior for the driving inputs and thus there is no way to validate that they are correct, however with an appropriate experiment (e.g. applying a perturbation or stimulus at a specific time) perhaps they could be useful.
>
> > Generally, more concretely linking any of the observations to real neural or behavioral variables would improve the manuscript’s impact.
>
> Thank you for highlighting the concern regarding limited application. To address this, we implemented our framework on a C. elegans dataset, capitalizing on a significant advantage — the presence of controlled external stimuli in certain trials. We exploited this well-timed laser heat shock as a ground truth control input. In **new Figure 4.D**, we show a salient control is inferred at the moment of the laser pulse, a feature not readily apparent in the posture space. This outcome is particularly encouraging as it suggests the potential for the model to be employed in conjunction with perturbation experiments in the future. For instance, it could be used alongside optogenetic stimulation applied to a population of neurons. In such experiments, the timing would be controlled by the experimentalist, while the impact on the dynamical system, though unknown, could be inferred by our model.
>
> >`Q1.` Does the number of control inputs match the number of bout types people have reported in the literature?
>
> This is an interesting point you raised. To clarify, we found that rather than controlling distinct movements, the different control inputs were combined in different ratios to generate different categories of movements. We added a figure illustrating this (**Figure S3**).

---

> > ### Comment · Reviewer_j5DC · 2023-11-23
> > **Thanks for the reply**
> >
> > Thanks for the reply. I do think these substantive changes directly address my questions which were shared by some of the reviewers eg AsPq  and 9AdGwho note the need to do a proper model comparison and 9AdG who notes the limited scope of the results for application papers where it is common to demonstrate general applications in a few use cases. I will increase my score based on these results.

---

### Official Review · Reviewer_325F · 2023-11-01

**Soundness:** 3 good
**Presentation:** 3 good
**Contribution:** 2 fair
**Rating:** 6
**Confidence:** 4

**Summary:**

In this paper the authors propose a control framework for estimating movement dynamics from behavioural/postural observations. Movements are considered as generated by an unknown dynamical system controlled through sparse input signals.
They employ two alternative methods to fit the latent dynamics:
- One that uses the iLQR-VAE from Schimel et al., 2022 that approximates the observed dynamical system through an RNN.
- And a linear model of “large dimensionality” that is consequently reduced to a lower dimensional model using balanced truncation.
They demonstrate their method on a toy example of a linearly approximated system comprising two pendulums, and on a behavioural dataset of zerbafish movements.
They are able to further provide insights into the identified dynamics by dissecting the spectrum of the reduced linear model.

**Strengths:**

- Applying balanced model reduction together with dynamical and control inference is an interesting contribution to the existing system identification literature.
- The linear approximation through the model reduction allows for interesting insights into the approximated dynamics (like in Figure 4 for analysing the eigenmodes active during different movements), that would be otherwise more cumbersome to perform for an RNN.

**Weaknesses:**

- The dissection of the identified dynamics through the reduced linear model requires the non-reduced linear system to already approximate accurately the observed system. I wonder whether the authors could provide a systematic analysis on the robustness of their framework when fit on the nonlinear toy model.
- As the authors already mention in their discussion, the framework they propose requires the fitting of two dynamical systems: an RNN that captures accurately the observed behavioural trajectories, and an linear system that provides interpretability.

**Questions:**

- In the toy example with the pendulum, as I understand, the authors created observations by simulating the linear approximation of the system (assuming small angles). I wonder how  the proposed framework would perform if the authors created the observations for the same system parameters with the nonlinear dynamics, and for parameter sets that result in increasingly larger angles. I think in this toy example it is interesting to demonstrate the robustness of the linear-reduced order framework.

- In Figure 3 why does the classification accuracy decrease with time (observation time?)?

- In Figure 4C and associated main text, the authors mention that the reduced model added eigenmodes with larger timescales, while it neglected the small timescale modes of the the non-reduced system. While the second part of the previous sentence is expected, I am not sure how should I understand the first one. Can you provide some intuition. Also related to this, when I first looked at Figure 4C I thought that the large the large time scale values of the reduced and non-reduced model overlap, therefore the light grey ones are not visible. Can you probably make the circles of the reduced system non-opaque or non-filled to make the plot clearer?

- I think it would be helpful if the authors mention in the supplement how they fit the linear model of their framework, and provide a brief description of the iLQR-VAE framework that is a crucial component of the proposed approach.

- Model Selection C.1. section is missing from the supplement.

- There is a typo in the subscripts in Eq. 5.

---

> ### Author Response · Authors · 2023-11-20
> **Response to 325F (1/2)**
>
> > Applying balanced model reduction together with dynamical and control inference is an interesting contribution to the existing system identification literature.
>
> >The linear approximation through the model reduction allows for interesting insights into the approximated dynamics (like in Figure 4 for analyzing the eigenmodes active during different movements), that would be otherwise more cumbersome to perform for an RNN.
>
> Thank you for your positive feedback on the insights from the model reduction, we also believe that this is a promising method to facilitate interpretability. This method was also successful when applied to the new C. elegans dataset, with the first 3 modes of the reduced model corresponding to reverse crawls, omega-turn, and forward crawls (see **new Figure S8**).
>
>
> > In the toy example with the pendulum, as I understand, the authors created observations by simulating the linear approximation of the system (assuming small angles). I wonder how the proposed framework would perform if the authors created the observations for the same system parameters with the nonlinear dynamics, and for parameter sets that result in increasingly larger angles. I think in this toy example it is interesting to demonstrate the robustness of the linear-reduced order framework.
>
> Thank you for raising this point! Indeed, we generated this toy example by assuming that the system was linear. The main point of this example was to illustrate how the model reduction works and act as a sanity check that iLQR-VAE + model reduction has the expected performance, in the case where ground truth is known.
>
> While we do agree that the example you propose is interesting, it is not obvious to us how it demonstrates the robustness of the framework: indeed, as we increase the angles, the model will not recover the linearized dynamical system, but the linearized system does not exactly capture the ground truth dynamics, so it is not obvious to us whether that is a failure of the model.
>
> In general, as the behavioral time series we are analyzing are unlikely to have true underlying linear dynamics, we view the linear + model reduction approach as a way to extract interpretable modes from the model rather than as a way to recover the ground truth. Here, our examples hopefully illustrate this – even though we know we are modeling nonlinear systems, the linear approximation + model reduction allows us to extract novel insights that are less accessible otherwise.
>
> We do however agree that the question of where the linear model may fail is interesting: in our view, this is better addressed by fitting both a nonlinear model and a linear model, and to evaluate which parts of the dynamics can be captured by the former but not the latter.
>
>
> > In Figure 3 why does the classification accuracy decrease with time (observation time?)?
>
> In Figure 3, a distinct classifier is trained at each time point (excluding the dashed line that represents training from the complete time series of posture). Therefore as time advances toward the end of the movements, the classification gradually diminishes, approaching the chance level.
>
> > In Figure 4C and associated main text the authors mention that the reduced model added eigenmodes with larger timescales, while it neglected the small timescale modes of the non-reduced system. While the second part of the previous sentence is expected, I am not sure how should I understand the first one. Can you provide some intuition. Also related to this, when I first looked at Figure 4C I thought that the large time scale values of the reduced and non-reduced model overlap, therefore the light grey ones are not visible. Can you probably make the circles of the reduced system non-opaque or non-filled to make the plot clearer?
>
> Thank you for this question. That is a good point, and indeed the behavior of the model reduction has interesting properties that are not all intuitive (which is another reason why we believe our example can be informative to other researchers in the area).
>
> The reduced model learns modes that differ slightly from the top eigenmodes of the previous system, as it is a different linear system. In the original system, it is possible for multiple modes to combine with one another, leading to effectively longer timescales arising in the system. Since the reduced model looks for a minimal version of the dynamics, it might learn timescales that do not exist in the initial system (and that may be longer). Let us know if this clarifies things. We have, in addition, followed your advice and **updated Figure 3c** with open circles for the reduced model spectrum.

---

> > ### Author Response · Authors · 2023-11-20
> > **Response to 325F (2/2)**
> >
> > > I think it would be helpful if the authors mention in the supplement how they fit the linear model of their framework, and provide a brief description of the iLQR-VAE framework that is a crucial component of the proposed approach.
> >
> > This is a good point, we have included this information in the form of supplementary sections. Briefly, here, we found the linear model by refitting a sparse linear RNN to the data.
> >
> > > Model Selection C.1. section is missing from the supplement. There is a typo in the subscripts in Eq. 5.
> >
> > Thank you for spotting those, we have amended the manuscript to correct them!

---

> ### Comment · Reviewer_325F · 2023-11-22
>
> I would like to thank the authors for their responses.
> I have read the responses to my comments and to the comments of the other reviewers, and I think our concerns have been mostly addressed.
>
> I appreciate that the authors put in the effort to additionally compare their method to LFADs, and included the additional eigenworm decomposition of C. elegance posture dynamics. This seems to be a more suitable application for their framework compared to the zebrafish dataset, since the experimental setup includes externally delivered perturbations.
>
> >The reduced model learns modes that differ slightly from the top eigenmodes of the previous system, as it is a different linear system. In the original system, it is possible for multiple modes to combine with one another, leading to effectively longer timescales arising in the system. Since the reduced model looks for a minimal version of the dynamics, it might learn timescales that do not exist in the initial system (and that may be longer). Let us know if this clarifies things. We have, in addition, followed your advice and updated Figure 3c with open circles for the reduced model spectrum.
>
> Thank you for the clarification. It makes sense that the reduced model might learn slightly different modes compared to the top eigenmodes of the high-dimensional linear system. However, I still have some concerns about the additional dominant modes introduced in the zebrafish experiment's reduction process.
>  I would expect from the reduced model to capture the dominant behaviour of the system, thus more or less its spectrum to contain the top eigenvalues of the high-dimensional linear system (akin to the reduced spectrum in the C. elegance experiment [Figure S8]), without adding additional *dominant* ones.
> I would be curious to see what kind of behaviour these dominant modes of the reduced model represent, and whether they would disappear if you would change the order of the reduction. But I won't insist on this.
>
>
>
> Minor:
> - Figure 2C,  legend key for the whole posture time series should be a dashed line.

---

> ### Author Response · Authors · 2023-11-23
> **2nd Response to 325F**
>
> Thank you very much for your positive feedback and for engaging in the rebuttal process, we very much appreciate it!
>
> > Thank you for the clarification. It makes sense that the reduced model might learn slightly different modes compared to the top eigenmodes of the high-dimensional linear system. However, I still have some concerns about the additional dominant modes introduced in the zebrafish experiment's reduction process. I would expect from the reduced model to capture the dominant behaviour of the system, thus more or less its spectrum to contain the top eigenvalues of the high-dimensional linear system (akin to the reduced spectrum in the C. elegance experiment [Figure S8]), without adding additional dominant ones. I would be curious to see what kind of behaviour these dominant modes of the reduced model represent, and whether they would disappear if you would change the order of the reduction. But I won't insist on this.
>
> This is a very interesting question, thank you for raising this point!
> Our hypothesis for this behavior was that the original linear dynamical system that the model learns is highly non-normal. This means that the connectivity contains hidden feedforward chains which can lead to the dynamics exhibiting timescales that are effectively longer than the slowest timescale in the eigenspectrum (see [`1`] for illustrations of such behaviors in recurrent neural networks).
>
> Our intuition was that the reduced models are less non-normal than the original dynamics matrix, and thus need to rely on slower eigenvalues to be able to generate the same slow timescales as the original system.
> We have now confirmed this by computing the eigenvalue spectrum at different orders of the reduction (see **[link](https://anonymous.4open.science/r/iclr_rebuttal_figs-F77E/Figure1.png) to Figure 1**), as well as the degree of nonnormality of the reduced system (**see [link](https://anonymous.4open.science/r/iclr_rebuttal_figs-F77E/Figure2.png) to Figure 2**). We computed the nonnormality of $A$ here as,
> $$ 1 - { \sqrt{ \sum_{i=1}^{r} \lambda_{i}^{2}} \over \vert\vert{A^{(r)}}\vert\vert^{2}_{2}} $$
> which is equivalent to computing the norm of the upper Schur triangle of A, divided by the norm of A (there, a fully non-normal matrix would have a degree of non-normality of 1, and a fully normal matrix would have a degree of 0).
>
> There indeed appears to be a very clear trend, with the reduced systems getting gradually more and more normal as we reduce the order of the system, and thus relying on slower timescales to approximate the dynamics of the original system.
> While there could also be other effects at play, we believe that this in large part explains why those slow eigenvalues arise in the reduced systems.
>
>
> > I have read the responses to my comments and to the comments of the other reviewers, and I think our concerns have been mostly addressed.
>
> We are glad that you feel we have addressed the main points raised by all the reviewers. There were several very interesting questions raised and we hope you think, as we do, that the new data, together with this additional analysis have significantly improved the paper.
>
> [`1`] Mark S Goldman, 2009. Memory without Feedback in a Neural Network. Neuron.

---

### Official Review · Reviewer_9AdG · 2023-11-01

**Soundness:** 3 good
**Presentation:** 3 good
**Contribution:** 2 fair
**Rating:** 5
**Confidence:** 4

**Summary:**

The paper proposes a framework to identify the latent control signals and the underlying dynamics that make up the natural behavior of zebrafish. It utilizes iLQR-VAE for learning the parameters and inferring the control signals. The paper trained recurrent neural network (RNN) models and linear dynamical systems (LDS) to reproduce the postural sequence with sparse control signals. The underlying dynamics of those models are explored and how they relate to the behaviors is studied. The paper further demonstrates the model order reduction on the LDS model and relates the reduced mode to the observed behaviors.

**Strengths:**

The paper looks at a new approach for studying naturalistic behaviors, and defines the problem in a clear way.

The model reduction on the LDS model gives new insights into modeling behavior.

**Weaknesses:**

The paper utilizes iLQR-VAE for learning the parameters and inferring the control signals. It would be helpful if the authors stated clearly in the text what is the difference between the proposed model and the previous model. It seems like the differences are minimal if any, in which case, this is a good application paper, however, it doesn't introduce many novel elements from a modeling perspective. While the model reduction technique for linear models is not usually applied to this field of behavioral modeling, it is a very well-known concept and does not offer novelty from a methodological perspective. There are interesting takeaways from a neuroscience perspective, but these would have to be validated more thoroughly and may be more suitable for a different venue.

The kinematics of the zebrafish are somewhat simple; however, the naturalistic behaviors of other animals (mice and monkeys, for example) are much more complex. Will the model still be able to capture these dynamics? One real-world dataset may not be able to provide enough insight about the generalizability of the model for this question.

There is no comparison with other models presented.


Minor:
1. Section 4.1, mismatched figure labels. Figure 1C->Figure 1B; Figure 1D -> Figure 1C
2. Figure 3 Caption: missing space: ‘udriving’ -> ‘u driving’
3. More explanations on Figure 3D in the main text.
4. Page 15: ‘all states withing the LDS’ -> ‘all states within the LDS’
5. Page 15: ‘cannot be reach via observations’ -> ‘cannot be reached via observations’
6. Page 16: ‘Therefore, we can the compute the’ -> ‘Therefore, we can compute the’

**Questions:**

Please see above weaknesses.

---

> ### Author Response · Authors · 2023-11-20
> **Response to 9AdG**
>
> > The model reduction on the LDS model gives new insights into modeling behavior.
>
> Thank you for noting this, we also think that the model reduction is an interesting framework for behavioral modeling. Our new C. elegans (see **new Figure 4**) results corroborate the fact that it can be a useful way to extract relevant modes of animal behavior.
>
> > It would be helpful if the authors stated clearly in the text what is the difference between the proposed model and the previous model. It seems like the differences are minimal if any, in which case, this is a good application paper, however, it doesn't introduce many novel elements from a modeling perspective.
>
> >  There are interesting takeaways from a neuroscience perspective, but these [...] may be more suitable for a different venue.
>
> Thank you for your comment: indeed, our paper focuses on the application of iLQR-VAE to naturalistic animal behavior. We fine-tuned the model parameters to work on the datasets of interest to us, but we indeed did not change the existing method.
>
> We would indeed describe our work as “*an application paper*”, more than proposing a new methodology. Note however that, **in the ICLR call for papers, “*applications in audio, speech, robotics, neuroscience,  biology, or any other field*” are considered and welcome. Therefore, we do not think that our focus on a neuroscience application should be ground for rejection.**
>
> >  While the model reduction technique for linear models is not usually applied to this field of behavioral modeling, it is a very well-known concept and does not offer novelty from a methodological perspective.
>
> We agree that the model reduction (MOR) approach is certainly not new. However, we are not aware of other work applying it to behavioral modeling or neuroscience. We thus believe our paper would be very relevant to members of those communities, as a case study using MOR to extract insights from data.
>
> Additionally, applying model reduction to an input-driven dynamical system whose parameters are learned from data is to the best of our knowledge a novel approach. Asides from iLQR-VAE and LFADS, no methods have focused on learning input-driven dynamics from data.  We believe that the combination of such approaches with MOR opens up very exciting avenues for the field of systems identification, and is a valuable contribution.
>
> > There is no comparison with other models presented.
>
> We agree that the lack of comparison with other methods was a weakness of the paper. We have now included a comparison with LFADS on the zebrafish dataset. In summary, we found that LFADS (which assumes an autoregressive prior over the inputs), can fit the behavioral time series but does so using continuous inputs which are strongly correlated to the posture (see **new Figure S6**). Moreover, LFADS leads to latent trajectories which are less predictive of the behavioral category (see **updated Figure 2.C**).
>
> > The kinematics of the zebrafish are somewhat simple; however, the naturalistic behaviors of other animals (mice and monkeys, for example) are much more complex. Will the model still be able to capture these dynamics? One real-world dataset may not be able to provide enough insight about the generalizability of the model for this question.
>
> We agree that providing only the zebrafish as an example application somewhat limited the scope of the paper. We have addressed this by including a new application (see **new Figure 4** and **Figure S8**) to behavioral recordings from crawling C. elegans (worm). While the posture is still low-dimensional, the movement dynamics are more complex than the zebrafish because of their continuous nature, and the fact that the worm is changing its behavior in response to unknown environmental stimuli at variable time intervals.
>
> We found that the model performed well on this dataset. It reconstructed the behavior accurately, using inputs that strongly correlated to ground truth aversive heat stimuli (**new Figure 4.D**).
>
> Additionally, after performing the model reduction on the learned dynamical system, the top 3 modes of the reduced system matched the previously identified behavioral modes of the worm [`1`]. We hope that these results will convince you of the general applicability of our method, and the insights it can yield!
>
> > There are interesting takeaways from a neuroscience perspective, but these would have to be validated more thoroughly
>
> As we detail above, we believe that our new C. elegans results address this concern, both because of the ground truth input recovery, and the fact that the model reduction identifies modes that can be related to prior work.
>
> **Regarding Minor Changes**
> Thank you for spotting the few typos and mislabelling of figures. We have made these amendments to the updated manuscript.
>
> [`1`] Costa, A.C., Ahamed, T. and Stephens, G.J., 2019. Adaptive, locally linear models of complex dynamics. Proceedings of the National Academy of Sciences, 116(5), pp.1501-1510.

---

### Official Review · Reviewer_AsPq · 2023-11-06

**Soundness:** 3 good
**Presentation:** 3 good
**Contribution:** 3 good
**Rating:** 6
**Confidence:** 4

**Summary:**

This paper uses an unsupervised method of discovering latent control signals from behavioural sequences, to discover behavioural motifs. The main methodological contribution is using a model reduction technique to gain interpretability insights into the inferred models. Overall I think this is a good paper.

**Strengths:**

This paper analyses various behavioural sequences and infers intuitive behavioural motifs.

While the core method is not new, the interpretability aspect is, and the insights are interesting.

The paper is well written, and clearly presented.

**Weaknesses:**

The paper could do with a proper model comparison, i.e. vs moseq.

The Balanced truncation bit needs to be better explained with some intuition.

Eqn 5 should have $ \tilde{W}_o = \tilde{W}_c = \dots$

Fig 1 A: what does blue and orange correspond to? Different sequences?

Some of the Figures seem to be mis-referenced, e.g., “As shown in Figure 3B, the MGU could reconstruct bouts using a sparser control compared to the LDS. "

All the figures could do with more explanations in their captions.

**Questions:**

See weaknesses

---

> ### Author Response · Authors · 2023-11-20
> **Response to AsPq**
>
> > This paper analyses various behavioral sequences and infers intuitive behavioral motifs.
>
> > While the core method is not new, the interpretability aspect is, and the insights are interesting.
>
> > The paper is well-written, and clearly presented.
>
> Thank you for your positive feedback! We also think the insights that can be obtained by applying model reduction are interesting. We would like to point you to the C. elegans results that we have added in the updated version of the paper (see **new Figure 4 and S8**), that provide another example of the insights that model reduction applied to a learned dynamical system can yield. Indeed, we find that, in this more complex C.elegans dataset, our method recovers most important modes of the system that are consistent with previously identified behavioural patterns [`1`] – something we were very excited about.
>
> >  The paper could do with a proper model comparison, i.e. vs moseq.
>
> Thank you for your comment. We agree that the original version of the paper was lacking a comparison with other methods. We have now added a comparison to LFADS (a method similar to iLQR-VAE, but with a different recognition model, and which typically uses an autoregressive input prior).  We found that LFADS can reconstruct the behavior, but using inputs that are highly correlated with the posture (see **new Figure S.6**). Additionally, it gives rise to latent trajectories that are a lot less predictive of the behavior than the MGU model (see **updated Figure 2.C**).
>
> Note that Moseq is indeed a widely used method in the field of behavioral modeling, but its principal use is for segmenting movements into motifs. Thus, since the zebrafish behavior is by nature already segmented in bouts, we did not think that MoSEQ would yield additional insights. However, if you had a specific comparison in mind that you would like us to do we would be happy to hear that!
>
> > The Balanced truncation bit needs to be better explained with some intuition.
>
> Thank you for that suggestion – we have tried our best to make the balanced truncation as intuitive as possible (see for instance the intuition given regarding the observability or controllability Gramians, or the intuition for the Hankel singular values).
>
> We have also moved the pendulum example to the Appendix, and included more details on how it illustrates an example case of the balanced truncation, hopefully providing further intuition for the behavior of the model.
>
> However, if you have any suggestions on how to further improve the clarity of this section (e.g. specific places that you find confusing) we would be happy to further amend the paper!
>
> > Fig 1 A: what does blue and orange correspond to? Different sequences?
>
> That is a good point – blue and orange correspond to two different input / output channels of the pendulum. We have clarified this in the caption.
>
> > typos and captions and misreferences
>
> Thank you very much for spotting those, we have amended the manuscript to fix them!
>
> [`1`] Costa, A.C., Ahamed, T. and Stephens, G.J., 2019. Adaptive, locally linear models of complex dynamics. Proceedings of the National Academy of Sciences, 116(5), pp.1501-1510

---

> > ### Comment · Reviewer_AsPq · 2023-11-23
> > **Thanks for the response**
> >
> > Thanks for the response and the additional result. I still like the paper, but will stick with my original score.

---

### Author Response · Authors · 2023-11-20
**Response to all reviewers**

First and foremost, we would like to thank all reviewers for their detailed and insightful reviews. We are encouraged that they found our motivation and idea to be *well formulated* (**`AsPq`**,**`9AdG`**,**`j5DC`**) and *novel* (**`j5DC`**), and our *model reduction analysis insightful* (**`AsPq`**,**`9AdG`**,**`j5DC`**, **`325F`**). And overall our work to be a *good paper* (**`AsPq`**) and, *good application paper* (**`9AdG`**). The feedback has greatly helped us improve the paper and potential avenues for future work.

Some concerns shared across all reviewers were the fact that we only applied our method to a single real-world dataset (**1.**), and that we did not show comparisons with alternative methods (**2.**).

We agree that those are very reasonable concerns, we addressed them by including:

1. An additional example to the paper, is in which we successfully applied the method to another animal, the C. elegans (nematode worm) [`1`].

The additional C. elegans results provide strong support for the general applicability of our approach. Indeed, the worm moves in a continuous manner (unlike the zebrafish which swims in discrete bouts). While this makes the modeling problem more challenging, we found that our method could successfully handle such continuous recordings, being able to reconstruct the worm’s behavior highly accurately, and inferring inputs to explain transitions in the behavior (e.g switches from forward crawls to reverse crawls and omega-turn, see **new Figure 4**).  Interestingly, we found those behaviors could be mapped to modes of the reduced model (see **new Figure S8**).  Finally, this dataset included ground truth laser inputs, which we found to be successfully recovered by the model (see **new Figure 4.D**).

2. A comparison with LFADS, the method that provides the most relevant comparison point in our opinion.

There, we found that the latent trajectories inferred by LFADS were much less predictive of the behavioral category than those found by using iLQR-VAE (see **updated Figure 2.C**) This is due to the fact that the autoregressive prior for the inputs in LFADS doesn’t promote a strong autonomous dynamical system (see **new Figure S6**).

Those results show that our method can indeed be applied to model other, arguably more complex behaviors. Moreover, they illustrate the unique aspects of our approach which make it particularly well-suited to behavioral modeling problems. We have incorporated those additional experiments in the paper, both in the form of a short additional section in the Results (final section, associated with the new Figure 4), and several Supplementary sections. We also corrected typos and clarified the text following all the Reviewers' suggestions.

In our individual responses, we will point to the changes we made to address more specific comments.

[`1`] Broekmans, Onno D.; Rodgers, Jarlath B.; Ryu, William S.; Stephens, Greg J. (2016). Data from: Resolving coiled shapes reveals new reorientation behaviors in C. elegans [Dataset]. Dryad. https://doi.org/10.5061/dryad.t0m6p

---

### Meta-Review · Area_Chair_at5E · 2023-12-05

**Metareview:**

The authors propose a control framework for estimating movement dynamics from behavioural/postural observations. Movements are considered as generated by an unknown dynamical system controlled through sparse input signals, which are inferred using the  iLQR-VAE from Schimel et al., 2022. The method is demonstrated both on a toy example which approximates a coupled system of two pendulums, and on a behavioural dataset of zebrafish movements.

While the paper represents a useful and clearly presented example of an application of an existing method to a new application domain, and is backed up by clear analysis, there was some concerns about the methodological novelty of the paper, lack of benchmark comparisons (although some were added during the rebuttal), and the reviewers did not see clear evidence that the approach enables fundamentally new types of analyses.

**Justification For Why Not Higher Score:**

See above-- all reviewers gave 5 or 6, and no-one argued strongly for (or, for that matter) against the paper.

**Justification For Why Not Lower Score:**

I, actually think this is a neat paper and there is nothing wrong with it,  but I agree that there is also nothing radically new-- the question is where one draws the line for 'existing enough'. If there is space at ICLR, it would be great if this could be presented, but I would not mind it being bumped down.

---

### Decision · Program_Chairs · 2024-01-16

Accept (poster)